# CLIP Tricks You: Training-free Token Pruning for Efficient Pixel Grounding in Large Vision-Language Models

**Sangin Lee** [1]    **Yukyung Choi** [1 2]

## Abstract

In large vision-language models, visual tokens typically constitute the majority of input tokens, leading to substantial computational overhead. To address this, recent studies have explored pruning redundant or less informative visual tokens for image understanding tasks. However, these methods struggle with pixel grounding tasks, where token importance is highly contingent on the input text. Through an in-depth analysis of CLIP, we observe that visual tokens within referent regions often exhibit low similarity to their textual representation. Motivated by this insight, we introduce LiteLVLM, a training-free, text-guided token pruning strategy for efficient pixel grounding inference. By reversing the ranking of CLIP's visual-text similarity, LiteLVLM effectively retains visual tokens covering the referent regions, while recovering context tokens to enable clear foreground-background separation. Extensive experiments demonstrate that LiteLVLM significantly outperforms existing methods by over 5% across diverse token budgets. Without any training or fine-tuning, LiteLVLM maintains 90% of the original performance with a 22% speedup and a 2.3× memory reduction. Our code is available at https://github.com/sejong-rcv/LiteLVLM.

## 1. Introduction

With recent advances in Large Language Models (LLMs) (Ouyang et al., 2022; Team et al., 2024; Bai et al., 2023), efforts to extend their reasoning capabilities to visual information have driven rapid progress in Large Vision-Language Models (LVLMs) (Achiam et al., 2023; Team et al., 2023; Bai et al., 2025a; Chen et al., 2024b). Beyond vision-

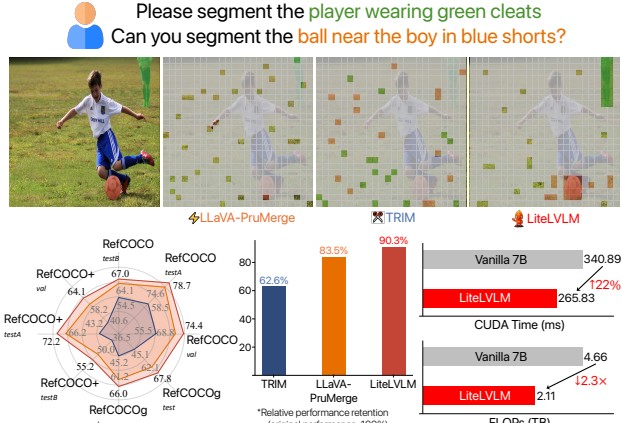

*Figure 1.* **Comparison of different token pruning methods. Top**: Colored patches indicate the retained visual tokens for each referring expression. Our LiteLVLM effectively preserves the tokens corresponding to the referent. **Bottom**: LiteLVLM achieves the best performance across all referring expression segmentation benchmarks, retaining around 90% performance with 66.7% token pruning. LiteLVLM also improves efficiency, with a 22% inference speedup and a 2.3× reduction in memory overhead.

language tasks, LVLMs have been extended to pixel grounding tasks (Liu et al., 2023), enabling region-level visual reasoning and interpretation. However, their inference is computationally expensive and time-consuming, primarily due to the processing of visual tokens within the LLM decoder. For instance, in LLaVA (Liu et al., 2024c), the vision encoder generates 576 visual tokens, while text tokens typically number fewer than 100. Visual token counts further increase when the model accepts higher-resolution images (2880 for LLaVA-NeXT (Liu et al., 2024b)) or video inputs (2560 for Video-LLaVA (Lin et al., 2024)). To make inference faster and more efficient, a natural question follows: *"Should all visual tokens be retained and used?"*

Motivated by this insight, several works (Xing et al., 2025; Li et al., 2025) have studied token pruning methods to reduce redundant or less informative visual tokens. To determine which tokens to retain, LLaVA-PruMerge (Shang et al., 2025) retains tokens based on similarities between visual tokens and the [CLS] token, which serves as a global representation in CLIP (Radford et al., 2021). Despite their success in image understanding tasks (Goyal et al., 2017; Li

---

[1]Dept. of Artificial Intelligence and Robotics, Sejong University, Seoul, Republic of Korea [2]Artificial Intelligence and Robotics Institute (AIRI), Sejong University, Seoul, Republic of Korea. Correspondence to: Yukyung Choi <ykchoi@sejong.ac.kr>.

*Proceedings of the 43rd International Conference on Machine Learning*, Seoul, South Korea. PMLR 306, 2026. Copyright 2026 by the author(s).

et al., 2023b), these methods have not yet been evaluated for pixel grounding, where we find that they struggle with tasks such as referring expression segmentation (Kazemzadeh et al., 2014). As illustrated in Figure 1, LLaVA-PruMerge fails to preserve visual tokens within the referent regions, as it solely relies on text-agnostic visual importance, leading to degraded performance. From this observation, we argue that token importance in pixel grounding varies with the input referring expression (*e.g.*, "player" and "ball"), necessitating a text-guided pruning strategy.

To this end, we conduct an in-depth analysis of CLIP and observe a **visual-text similarity reversal.** Interestingly, we find that visual tokens located within referent regions exhibit lower similarity to the referring expression. In Figure 1, TRIM (Song et al., 2025) selects visual tokens with high visual-text similarity, which are often irrelevant to the referent, resulting in a severe performance degradation. In contrast, retained tokens with low visual-text similarity (see Figure 1-LiteLVLM) are spatially well-aligned with the referent, preserving rich cues for grounding the object. We suggest that this phenomenon is a natural consequence of CLIP's contrastive pretraining. To better understand this finding, we investigate text tokens in CLIP and reveal a **text attention sink**. Specifically, we observe that the attention of the [EOS] token is heavily biased toward the special token, *i.e.*, the start-of-sentence ([SOS]) token (Chefer et al., 2023; Yi et al., 2024). Consequently, the lack of rich semantics in the [EOS] token fuels the similarity reversal, making high-similarity visual tokens irrelevant to the referent. Although these phenomena generally do not affect LVLMs during full-token inference, they lead to a substantial performance drop when pruning tokens based on textual information.

In this paper, we propose **LiteLVLM**, a simple yet effective training-free token pruning method that utilizes reversed CLIP visual-text similarity for efficient pixel grounding inference. LiteLVLM first retains similarity-aware tokens with low similarity to the [EOS] token for each input text. To preserve the global context, we then recover tokens based on high [CLS] contribution scores. Moreover, we propose an adaptive token selection strategy that dynamically adjusts the budget of both token types. To validate our method, we re-implement existing token pruning methods and conduct a comparative evaluation on pixel grounding tasks across image and video benchmarks. Extensive experimental results demonstrate that our LiteLVLM achieves state-of-the-art performance on pixel grounding tasks while significantly reducing the inference overhead.

Our contributions can be summarized as threefold:

1. We find that existing text-agnostic token pruning methods relying on the global importance of visual tokens perform poorly on pixel grounding tasks, highlighting the necessity of a text-guided pruning strategy.

2. Through a close analysis of CLIP, we observe that the [EOS] token overemphasizes the [SOS] token. Consequently, visual tokens within referent regions exhibit unexpectedly low similarity to the [EOS] token.

3. We introduce LiteLVLM, a text-guided token pruning method. To the best of our knowledge, it is the first to explore token pruning for pixel grounding. Extensive experiments validate that LiteLVLM reduces computational cost while maintaining superior performance.

## 2. Related Work

**Large Vision-Language Models (LVLMs).** Following the advent of Large Language Models (LLMs) such as LLaMA (Touvron et al., 2023), GPT (Achiam et al., 2023), and Gemma (Team et al., 2024), efforts to extend their reasoning abilities to vision tasks have led to remarkable advances in Large Vision-Language Models (LVLMs) (Liu et al., 2024c; Zhu et al., 2024; Team et al., 2023). LVLMs generally consist of a vision encoder, a vision-language projection layer, and an LLM. The vision encoder (*e.g.*, CLIP, SigLIP (Zhai et al., 2023)) encodes an image into a sequence of visual tokens. Then, the projection layer (*e.g.*, MLP, Q-former (Li et al., 2023a)) connects visual token representations into the word embedding space, enabling the LLM to process input token sequences and generate answers. However, while the number of text tokens is fewer than 100, the substantially larger number of visual tokens inevitably increases the computational burden. For example, LLaVA (Liu et al., 2024a) inputs a $336 \times 336$ image and generates 576 visual tokens. To process high-resolution images, Monkey (Li et al., 2024) encodes a $1344 \times 892$ image into 1792 tokens, and LLaVA-NeXT (Liu et al., 2024b) converts a $672 \times 672$ image into 2880 tokens. This issue becomes more pronounced in video understanding, where thousands to tens of thousands of visual tokens are processed (Lin et al., 2024; Kondratyuk et al., 2023). Therefore, reducing computational overhead and optimizing inference are urgent challenges for deploying LVLMs in resource-limited devices and applications.

**Token Pruning for LVLMs.** A straightforward way to improve LVLMs' efficiency is to prune visual tokens, which constitute the majority of the input sequence. Previous studies have primarily focused on pruning redundant or less informative visual tokens. These methods can be broadly categorized based on two criteria: where pruning is applied and whether textual information is used. When applied within the vision encoder (Shang et al., 2025; Zhang et al., 2025a; Yang et al., 2025), token pruning shortens the input sequence before entering the LLM, whereas pruning within the language model (Zhang et al., 2025b; Chen et al., 2024a; Lin et al., 2025; Ye et al., 2025) progressively removes tokens during the decoding process. Of these works, only TRIM (Song et al., 2025) leverages textual informa-

tion within the vision encoder by sparsifying tokens with low similarity to the CLIP text embedding. However, we observe that these discarded low-similarity tokens are often located within referent regions; thus, pruning them can considerably degrade grounding performance. Conversely, text-agnostic methods rely solely on visual information and thus struggle to identify tokens relevant to the text-specified referent. Moreover, these existing methods have been studied on image understanding tasks and not evaluated on pixel grounding. To the best of our knowledge, this is the first work extending token pruning to pixel grounding across image and video modalities.

## 3. Revisiting CLIP

In this section, we revisit CLIP by conducting an in-depth analysis. We begin in Section 3.1 by reviewing the contrastive learning objective of CLIP. Then, in Section 3.2 and Section 3.3, we analyze the visual-text similarity and reveal that CLIP text token excessively attends to the [SOS] token.

### 3.1. Preliminaries

We investigate CLIP (Radford et al., 2021), which is widely used as the vision encoder in LVLMs, to analyze the similarity between visual tokens and a text token. To this end, we start by reviewing CLIP's contrastive learning objective. CLIP is pretrained on millions of web-scale image-text pairs. Specifically, given a set of <image, text> pairs $\mathcal{S} = \{(\mathcal{I}_i, \mathcal{T}_i)\}_{i=1}^N$, a vision encoder and a text encoder embed the $i$-th image and text into global feature embeddings—the [CLS] and [EOS] token embeddings—denoted as $E_i^{\mathcal{I}}$ and $E_i^{\mathcal{T}}$, respectively. Then, contrastive learning aligns paired embeddings by pulling them closer while pushing unpaired ones apart. Formally, the contrastive loss is the sum of image-to-text loss $\mathcal{L}_{\mathcal{I} \to \mathcal{T}}$ and text-to-image loss $\mathcal{L}_{\mathcal{T} \to \mathcal{I}}$, which maximizes the cosine similarity of paired embeddings while minimizing that of unpaired ones:

$$\mathcal{L}_{\text{CLIP}} = \mathcal{L}_{\mathcal{I} \to \mathcal{T}} + \mathcal{L}_{\mathcal{T} \to \mathcal{I}}$$
$$= -\frac{1}{2N} \sum_{i=1}^N \log \frac{\exp(\cos(E_i^{\mathcal{I}}, E_i^{\mathcal{T}})/\tau)}{\sum_{j=1}^N \exp(\cos(E_i^{\mathcal{I}}, E_j^{\mathcal{T}})/\tau)} \quad (1)$$
$$- \frac{1}{2N} \sum_{j=1}^N \log \frac{\exp(\cos(E_j^{\mathcal{I}}, E_j^{\mathcal{T}})/\tau)}{\sum_{k=1}^N \exp(\cos(E_k^{\mathcal{I}}, E_j^{\mathcal{T}})/\tau)},$$

where $\cos(\cdot, \cdot)$ denotes the cosine similarity, $N$ is the mini-batch size, and $\tau$ is a learnable temperature parameter.

### 3.2. Visual-Text Similarity Reversal

To conduct a token-level analysis of visual-text similarity, we extract 576 visual tokens and an [EOS] token from CLIP ViT-L/14 on the RefCOCO-*val* subset (Kazemzadeh et al., 2014). Specifically, for the $i$-th image, the [CLS] token is

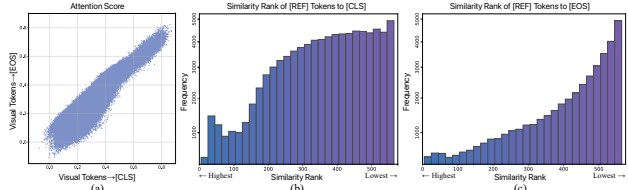

*Figure 2.* **Analysis of visual-text similarity.** (a) Attention correlation between [CLS] and [EOS]. Visual tokens show a clear positive attention correlation. (b) [REF]-[CLS] similarity rank distribution. (c) [REF]-[EOS] similarity rank distribution. [REF] tokens show even lower similarity to the [EOS] token than to the [CLS] token.

iteratively updated at each self-attention layer through an attention-weighted sum of visual tokens:

$$E_i^{\mathcal{I}} = \sum_{j=0}^M \alpha_{0,j} V_j, \quad \alpha_{0,j} = \text{softmax}\left(\frac{Q^{v\top} K_j}{\sqrt{d_k}}\right), \quad (2)$$

where $M$ is the number of visual tokens, and $\alpha_{0,j}$ is the attention weight between the [CLS] token and the $j$-th visual token $v_j$. $Q^v$ denotes the Query of the [CLS] token, while $K_j$ and $V_j$ denote the Key and Value of $v_j$, respectively. During contrastive learning, gradients from the image-to-text loss are backpropagated to the [CLS] token embedding $E_i^{\mathcal{I}}$ based on its cosine similarity with the [EOS] token embedding $E_i^{\mathcal{T}}$. Following the chain rule, these gradients are further backpropagated to all visual tokens, scaled by their attention weight $\alpha_{0,j}$. To intuitively inspect the gradient flow, we approximate the chain rule derivation as follows:

$$\frac{\partial \mathcal{L}_{\mathcal{I} \to \mathcal{T}}}{\partial v_j} = \frac{\partial \mathcal{L}_{\mathcal{I} \to \mathcal{T}}}{\partial E_i^{\mathcal{I}}} \cdot \frac{\partial E_i^{\mathcal{I}}}{\partial v_j}$$
$$\approx \frac{\partial \mathcal{L}_{\mathcal{I} \to \mathcal{T}}}{\partial \cos(E_i^{\mathcal{I}}, E_i^{\mathcal{T}})} \cdot \frac{\partial \cos(E_i^{\mathcal{I}}, E_i^{\mathcal{T}})}{\partial E_i^{\mathcal{I}}} \cdot \alpha_{0,j}. \quad (3)$$

Consequently, in Figure 2-(a), visual tokens with high attention to [CLS] also exhibit high attention toward [EOS], which aligns their representations and makes them similar.

Building on this finding, we next focus on the visual tokens located within referent regions. To this end, we identify tokens that overlap with the ground-truth segmentation mask (over 50%) and refer to them as [REF] tokens. These [REF] tokens encode essential information for grounding the object specified by the input text. In Figure 2-(b) and (c), we rank each [REF] token among all 576 visual tokens by its similarity to the [CLS] and [EOS] token, respectively. A rank closer to 0 indicates higher similarity, whereas a rank closer to 576 indicates lower similarity. Notably, the [REF] tokens show low similarity to the [CLS] token and even lower similarity to the [EOS] token, with their ranks concentrated near the bottom. These findings indicate that [REF] tokens are weakly aligned with global representations. Based on these observations, we argue that the **visual-text similarity reversal** stems from CLIP's contrastive pretraining, which

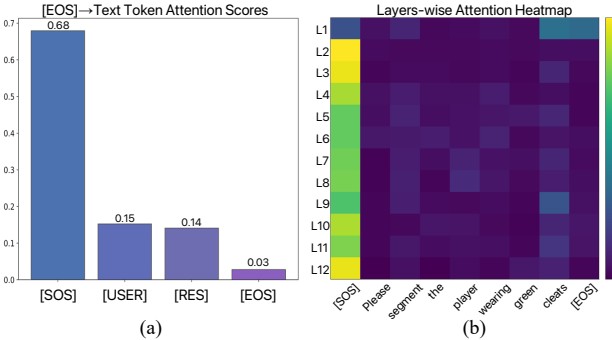

*Figure 3.* **Analysis of text attention sink.** (a) Average attention scores from the [EOS] to each text token. (b) Layer-wise self-attention heatmap from the [EOS] token to each word, where brighter colors indicate higher attention scores.

aligns feature embeddings of only the two global tokens. As a result, [REF] tokens receive weaker gradient signals from them, leading to lower similarity with global representations while preserving more localized, object-specific details.

### 3.3. Text Attention Sink

To deepen our understanding of the visual-text similarity reversal, we investigate the [EOS] token of CLIP. Typically, CLIP introduces two special tokens to explicitly bracket the sentence: the [SOS] token marks the beginning, while the [EOS] token not only marks the end but also encapsulates the entire text. Accordingly, the [EOS] token summarizes the semantic information from all text tokens into a single token. In Figure 3, we visualize the attention scores of the [EOS] token attending to other text tokens in the final transformer layer, averaged over the RefCOCO-*val* split. First, we automatically categorize text tokens into [USER] and [RES] groups using the NLP tool (Honnibal, 2017). Specifically, [USER] consists of tokens that match the user instruction, whereas [RES] represents the referring expression. For example, in the prompt "Please segment the player wearing green cleats", we assign "Please" and "segment" to [USER], and the referring tokens to [RES] (*e.g.*, "player", "cleats"). Since these [RES] tokens carry key information, we expect the [EOS] token to attend more strongly to them. However, as shown in Figure 3-(a), the [EOS] token concentrates most of its attention on the [SOS] token (0.68). Meanwhile, attention to [RES] tokens accounts for only 0.14, which is even slightly lower than that for [USER] tokens (0.15). Likewise, the layer-wise self-attention heatmap in Figure 3-(b) reveals that the [EOS] token shows a strong bias toward [SOS] beyond the first layer, while other tokens receive marginal attention. We attribute this bias to the fixed position of [SOS] token; in contrast, the positions of [REF] tokens vary across texts due to differing sequence lengths and padding. This **text attention sink** phenomenon thus hinders the [EOS] token from capturing semantic information, which further accelerates the visual-text similarity reversal.

## 4. Method

Based on our analysis of CLIP, we design **LiteLVLM**, a training-free token pruning method that leverages CLIP visual-text similarity for efficient pixel grounding. We first describe the overall architecture in Section 4.1. Then, we detail our method in Section 4.2 and Section 4.3, which selects the most informative tokens for grounding. Our method is plug-and-play and maintains strong performance even after reducing a significant number of visual tokens.

### 4.1. Architecture

LVLMs typically generate adequate responses for user-specified bounding-box regions but fall short in detailed pixel-level grounding. To address this, we employ GLaMM (Rasheed et al., 2024), which utilizes LLaVA-1.5 by linking LVLM output prompts to a pixel decoder for pixel grounding. Figure 4 provides an overview of LiteLVLM, which is built upon the GLaMM framework. Concretely, we leverage CLIP ViT-L/14 to encode the image $\mathcal{I}$ and the Vicuna-7B (Zheng et al., 2023) to encode the text $\mathcal{T}$. LiteLVLM prunes the visual tokens and forwards only the selected tokens $Z'_v$ through the projection layer $W$. Then, the language model integrates the projected visual features $H_v$ and the text features $H_q$ to generate the LVLM output. The language-to-prompt (L-P) projection layer transforms the LLM's last-layer embeddings of the specialized <SEG> token into the pixel decoder's feature space. The <SEG> token is used to specify the phrase to be grounded. Given the prompt "Can you segment the object?", the model generates <p>object</p><SEG> to specify "object" as the grounding target. The grounding image encoder—a pretrained SAM encoder (Kirillov et al., 2023)—extracts pixel-level embeddings. Finally, the SAM-style pixel decoder takes the <SEG> token embedding and pixel-level embedding to output the binary segmentation mask.

### 4.2. Similarity-aware Visual Token Selection

To determine which visual tokens to retain, it is necessary to identify tokens that best preserve model performance. Unlike image understanding tasks, which mainly rely on globally informative visual tokens, in pixel grounding, tokens corresponding to the referents are particularly important (*e.g.*, [REF] tokens). Considering the visual-text similarity reversal phenomenon, a simple and intuitive idea is to select visual tokens with low similarity to the [EOS] token $Z_{[EOS]}$. Although these tokens are weakly aligned with global semantics, they retain informative local cues about the referent and help maintain performance.

Given an image $\mathcal{I}$ and a set of $N$ input texts $\{\mathcal{T}_i\}_{i=1}^N$, we obtain each [EOS] token embedding $E_i^{\mathcal{T}}$ from the CLIP text encoder. Specifically, this embedding is extracted from

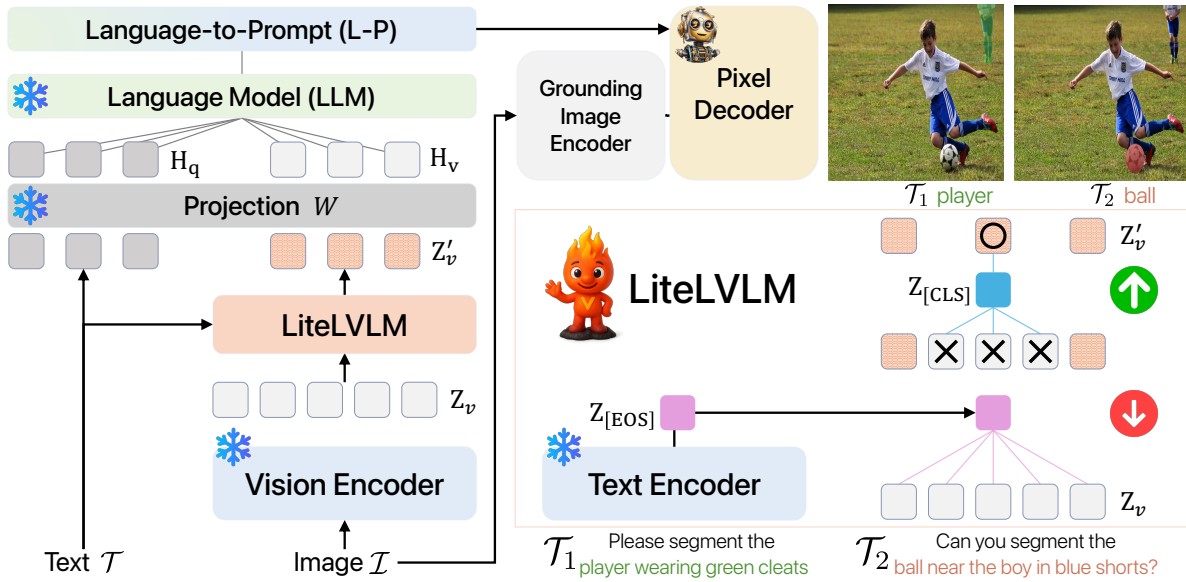

*Figure 4.* **Overview of LiteLVLM.** Given an image $\mathcal{I}$ and $N$ texts $\{\mathcal{T}_i\}_{i=1}^N$, we extract visual tokens $Z_v$ and [EOS] text token $Z_{\text{[EOS]}}$ from their respective encoders. We retain visual tokens with low similarity to $Z_{\text{[EOS]}}$, then recover contextually informative tokens using [CLS] token ($Z_{\text{[CLS]}}$) attention. The retained tokens ($Z_v'$) and text tokens are fed into the LLM and pixel decoder to generate a segmentation mask.

the last-layer hidden state and projected through its text-projection layer. Then, we compute the dot-product similarity scores $s_{i,j}$ between $E_i^{\mathcal{T}}$ and each $j$-th CLIP-projected visual token embedding $E_j^v$ as follows:

$$s_{i,j} = E_i^{\mathcal{T}} \cdot E_j^{v\top}, \quad j = 1, \dots, M. \tag{4}$$

Finally, we select the $k$ visual tokens with the lowest similarity scores, which tend to be located within referent regions.

### 4.3. Context-aware Visual Token Recovery

Since the similarity-aware tokens retained earlier primarily encode local information, pruning all context-rich tokens leads to a loss of semantic and background context. To remedy this, we recover contextually informative tokens by measuring each visual token's contribution to the [CLS] token $Z_{\text{[CLS]}}$. Specifically, for each $i$-th visual token in the non-retained token set $S$, we compute a contextual importance score $s_i'$ as the $L_2$ norm of the Value vector $V_i$ weighted by the dot-product attention between the [CLS] Query $Q^{\mathcal{I}}$ and the visual token key $K_i$, formulated as:

$$s_i' = \left\| \frac{\exp(Q^{\mathcal{I}} K_i^\top / \sqrt{d})}{\sum_{j \in S} \exp(Q^{\mathcal{I}} K_j^\top / \sqrt{d})} \cdot V_i \right\|_2. \tag{5}$$

Then, we recover the top-$k$ visual tokens with the highest scores. These context-aware tokens facilitate clearer boundary decisions while suppressing pixels outside the referent regions, leading to more precise pixel-level grounding. Together with the similarity-aware tokens, these tokens are fed into the LLM and transformed into the <SEG> grounding embeddings through the language-to-prompt layer. Then,

the pixel decoder combines these embeddings with dense image features to generate the pixel-level mask.

### 4.4. Adaptive Token Selection

To determine the number of similarity-aware and context-aware tokens, we design an adaptive token selection strategy. For a token budget $\mathcal{B}$, we first identify similarity-aware tokens by computing the visual-text similarity scores $s_{i,j}$ between the $i$-th input text and the $j$-th visual token. Then, we select low-similarity tokens for each text up to a budget $\mathcal{B}$ to yield $N$ candidate token sets $\mathcal{S}_i$, retaining only their intersection $\mathcal{S} = \bigcap_{i=1}^N \mathcal{S}_i$. For $N = 1$, given only a single candidate set, we empirically set the number of similarity-aware tokens in $\mathcal{S}$ to 50% of the budget. Among the remaining non-selected tokens, we recover top-scoring context-aware tokens using the context-aware scores $s_j'$ to fill the remaining budget ($\mathcal{B} - |\mathcal{S}|$). This dynamically balances both token types to adapt to diverse text inputs.

## 5. Experiments

In this section, we validate our LiteLVLM on various pixel grounding tasks across image and video modalities. We compare our LiteLVLM with existing methods and conduct ablation studies to analyze its effectiveness and efficiency.

### 5.1. Experimental Setup

**Evaluation Tasks.** We evaluate our method on three common referring expression segmentation datasets: RefCOCO, RefCOCO+, and RefCOCOg (Kazemzadeh et al., 2014;

*Table 1.* **Performance comparison of LiteLVLM on Referring Expression Segmentation.** The default number of visual tokens is 576. The first row reports the benchmark results, and the second row shows their proportion relative to the upper bound. Here, **Avg.** denotes the average score across 8 subsets, and **Rel.** indicates the average percentage of performance maintained at each reduction ratio.

| Method | RefCOCO | | | RefCOCO+ | | | RefCOCOg | | Avg. | Rel. |
|---|---|---|---|---|---|---|---|---|---|---|
| | val | testA | testB | val | testA | testB | val | test | | |
| *Upper Bound, All 576 Tokens* **(100%)** | | | | | | | | | | |
| GLaMM (CVPR24) | 79.5 | 83.2 | 76.9 | 72.6 | 78.7 | 64.6 | 74.2 | 74.9 | 75.5 | 100% |
| | 100% | 100% | 100% | 100% | 100% | 100% | 100% | 100% | | |
| *Retain 192 Tokens* (↓ **66.7%**) | | | | | | | | | | |
| TRIM (COLING25) | 55.5 | 58.5 | 54.5 | 40.6 | 43.2 | 36.5 | 45.2 | 45.1 | 47.3 | 62.6% |
| | 69.8% | 70.3% | 70.8% | 55.9% | 54.8% | 56.5% | 60.9% | 60.2% | | |
| FastV (ECCV24) | 69.5 | 73.9 | 63.6 | 57.9 | 64.2 | 48.3 | 61.5 | 62.0 | 62.6 | 82.9% |
| | 87.4% | 88.8% | 82.7% | 79.7% | 81.5% | 74.7% | 82.8% | 82.7% | | |
| LLaVA-PruMerge (ICCV25) | 68.8 | 74.6 | 64.1 | 58.2 | 66.2 | 50.0 | 61.2 | 62.1 | 63.1 | 83.5% |
| | 86.5% | 89.6% | 83.3% | 80.1% | 84.1% | 77.3% | 82.4% | 82.9% | | |
| VisionZip (CVPR25) | 71.1 | 76.4 | 66.6 | 59.7 | 67.2 | 54.0 | 64.7 | 64.3 | 65.5 | 86.7% |
| | 89.4% | 91.8% | 86.6% | 82.2% | 85.3% | 83.5% | 87.1% | 85.8% | | |
| VisPruner (ICCV25) | 72.4 | 75.5 | 66.8 | 61.5 | 66.9 | 54.2 | 65.4 | 65.2 | 66.0 | 87.4% |
| | 91.0% | 90.7% | 86.8% | 84.7% | 85.0% | 83.9% | 88.1% | 87.0% | | |
| **LiteLVLM** (ICML26) | **74.4** | **78.7** | **67.0** | **64.1** | **72.2** | **55.2** | **66.0** | **67.8** | **68.1** | **90.3%** |
| | 93.6% | 94.6% | 87.1% | 88.8% | 91.7% | 85.4% | 88.9% | 90.5% | | |
| *Retain 128 Tokens* (↓ **77.8%**) | | | | | | | | | | |
| TRIM (COLING25) | 53.1 | 54.4 | 50.0 | 37.1 | 39.9 | 35.4 | 42.8 | 42.8 | 44.4 | 58.8% |
| | 66.7% | 65.3% | 65.0% | 51.1% | 50.6% | 54.7% | 57.6% | 57.1% | | |
| FastV (ECCV24) | 64.9 | 69.9 | 58.4 | 51.9 | 59.3 | 43.2 | 56.6 | 56.8 | 57.6 | 76.3% |
| | 81.6% | 84.0% | 75.9% | 71.4% | 75.3% | 66.8% | 76.2% | 75.8% | | |
| LLaVA-PruMerge (ICCV25) | 64.2 | 70.9 | 60.3 | 54.1 | 61.0 | 47.1 | 57.6 | 58.8 | 59.2 | 78.4% |
| | 80.7% | 85.2% | 78.4% | 74.5% | 77.5% | 72.9% | 77.6% | 78.5% | | |
| VisionZip (CVPR25) | 66.4 | 71.1 | 60.9 | 54.5 | 62.2 | 47.9 | 59.6 | 59.0 | 60.2 | 79.7% |
| | 83.5% | 85.4% | 79.1% | 75.0% | 79.0% | 74.1% | 80.3% | 78.7% | | |
| VisPruner (ICCV25) | 66.7 | 72.5 | 64.0 | 55.8 | 62.2 | 48.6 | 59.6 | 59.8 | 61.1 | 80.9% |
| | 83.8% | 87.1% | 83.2% | 76.8% | 79.0% | 75.2% | 80.3% | 79.8% | | |
| **LiteLVLM** (ICML26) | **72.1** | **77.5** | **64.5** | **61.7** | **69.0** | **52.0** | **63.3** | **63.7** | **65.5** | **86.8%** |
| | 90.7% | 93.1% | 83.9% | 85.0% | 87.7% | 80.5% | 85.3% | 85.0% | | |
| *Retain 64 Tokens* (↓ **88.9%**) | | | | | | | | | | |
| TRIM (COLING25) | 50.1 | 52.6 | 49.2 | 33.1 | 35.4 | 30.9 | 38.7 | 39.4 | 41.1 | 54.1% |
| | 63.0% | 63.2% | 63.9% | 45.5% | 44.9% | 47.8% | 52.1% | 52.6% | | |
| FastV (ECCV24) | 57.3 | 60.8 | 52.4 | 42.1 | 45.1 | 37.6 | 47.9 | 48.6 | 48.9 | 64.8% |
| | 72.0% | 73.0% | 68.1% | 57.9% | 57.3% | 58.2% | 64.5% | 64.8% | | |
| LLaVA-PruMerge (ICCV25) | 58.9 | 64.3 | 54.9 | 45.9 | 50.3 | 42.4 | 49.5 | 50.0 | 52.0 | 68.8% |
| | 74.0% | 77.2% | 71.3% | 63.2% | 63.9% | 65.6% | 66.7% | 66.7% | | |
| VisionZip (CVPR25) | 59.0 | 63.8 | 55.1 | 47.1 | 51.9 | 40.1 | 49.2 | 51.7 | 52.2 | 69.1% |
| | 74.2% | 76.6% | 71.6% | 64.8% | 65.9% | 62.0% | 66.3% | 69.0% | | |
| VisPruner (ICCV25) | 57.8 | 62.7 | 54.3 | 45.7 | 49.3 | 40.5 | 49.8 | 52.3 | 51.5 | 68.2% |
| | 72.7% | 75.3% | 70.6% | 62.9% | 62.6% | 62.6% | 67.1% | 69.8% | | |
| **LiteLVLM** (ICML26) | **66.3** | **74.5** | **58.2** | **56.2** | **64.0** | **46.7** | **56.1** | **56.5** | **59.8** | **79.2%** |
| | 83.4% | 89.5% | 75.7% | 77.4% | 81.3% | 72.3% | 75.6% | 75.4% | | |

Mao et al., 2016). We also extend our evaluation to the video modality using two referring video object segmentation datasets: Ref-DAVIS-17 (Khoreva et al., 2018) and Refer-YouTube-VOS (Seo et al., 2020). Due to space limitations, the dataset details are provided in Section A.

**Token Budgets.** We set 576 visual tokens as the upper bound (100%). For referring expression segmentation, we evaluate performance under reduced token budgets of 192, 128, and 64, following the configurations of FastV (Chen et al., 2024a). For referring video object segmentation, we evaluate budgets of 196 and 81 tokens.

**Comparison Methods.** We adopt TRIM, FastV, LLaVA-PruMerge, VisionZip, SparseVLM, and the state-of-the-art VisPruner as comparison methods.

### 5.2. Referring Expression Segmentation

**Implementation Details.** To verify LiteLVLM, we employ GLaMM and fine-tune it on referring expression segmentation datasets. For a fair comparison, we re-implement the baselines within this same setup. Following prior work (Lai et al., 2024), we adopt cIoU—the cumulative intersection over the cumulative union—as our evaluation metric.

*Table 2.* **Performance comparison on Ref-DAVIS-17.**

| Model | $\mathcal{J}$ | $\mathcal{F}$ | $\mathcal{J}\&\mathcal{F}$ | Rel. |
|---|---|---|---|---|
| *Upper Bound, All 576 Tokens* **(100%)** | | | | |
| VideoGLaMM | 65.6 | 73.3 | 69.5 | 100% |
| *Retain 196 Tokens* (↓ 65.9%) | | | | |
| VisPruner | 61.2 | 67.4 | 64.3 | 92.5% |
| **LiteLVLM** | **66.8** | **71.6** | **69.2** | **99.5%** |
| *Retain 81 Tokens* (↓ 85.9%) | | | | |
| VisPruner | 57.1 | 63.5 | 60.3 | 86.7% |
| **LiteLVLM** | **64.3** | **67.8** | **66.1** | **95.1%** |

*Table 3.* **Performance comparison on Refer-YouTube-VOS.**

| Model | $\mathcal{J}$ | $\mathcal{F}$ | $\mathcal{J}\&\mathcal{F}$ | Rel. |
|---|---|---|---|---|
| *Upper Bound, All 576 Tokens* **(100%)** | | | | |
| VideoGLaMM | 65.4 | 68.2 | 66.8 | 100% |
| *Retain 196 Tokens* (↓ 65.9%) | | | | |
| VisPruner | 60.6 | 63.9 | 62.3 | 93.2% |
| **LiteLVLM** | **65.1** | **67.9** | **66.5** | **99.5%** |
| *Retain 81 Tokens* (↓ 85.9%) | | | | |
| VisPruner | 58.1 | 62.2 | 60.1 | 89.9% |
| **LiteLVLM** | **60.8** | **67.6** | **64.2** | **96.1%** |

*Table 4.* **Ablations for LiteLVLM.** All results are evaluated on RefCOCO-*val* under each retained token configuration. In the Similarity-aware Token Selection column, ↑ and ↓ denote selecting tokens with higher and lower visual-text similarity, respectively.

| #ID | Similarity-aware Token Selection | Context-aware Token Recover | Retain Tokens | | | | Avg. |
|---|---|---|---|---|---|---|---|
| | | | 29 | 58 | 144 | 288 | |
| 0 | ✗ | ✗ | 57.2 | 62.7 | 68.2 | 72.0 | 65.0 |
| 1 | ✗ | ✓ | 54.5 | 59.9 | 67.4 | 71.6 | 63.3 |
| 2 | ↑ | ✗ | 48.1 | 49.4 | 53.8 | 59.5 | 52.7 |
| 3 | ↓ | ✗ | 59.8 | 63.6 | 68.4 | 72.8 | 66.1 |
| 4 | ↓ | ✓ | **62.5** | **65.2** | **72.9** | **75.2** | **68.9** |

*Table 5.* **Ablations for token selection setup.** We analyze the impact of different ratios between similarity-aware token selection and context-aware token recovery on the RefCOCO-*val*. The last row represents our adaptive token selection setup.

| Similarity-aware Token Selection | Context-aware Token Recover | Retain Tokens | | | Avg. |
|---|---|---|---|---|---|
| | | 64 | 128 | 192 | |
| 25% | 75% | 63.5 | 68.5 | 70.9 | 67.6 |
| 50% | 50% | 63.7 | 69.9 | 73.3 | 68.9 |
| 75% | 25% | 62.4 | 69.1 | 72.7 | 68.1 |
| Adaptive Token Selection | | **66.3** | **72.1** | **74.4** | **70.9** |

**Main Results.** In Table 1, we present the performance of LiteLVLM on referring expression segmentation benchmarks. For a comprehensive analysis, we also report the average performance (Avg.) and the relative performance retention (Rel.), where vanilla GLaMM serves as the 100% upper bound. Notably, **LiteLVLM consistently outperforms existing methods across all benchmarks and token budgets** (192, 128, and 64 tokens). When reducing tokens from 576 to 192, LiteLVLM maintains performance with only a 9.7% drop in the average score, whereas FastV and VisPruner exhibit larger drops of 17.6% and 12.6%, respectively. Even with only 64 tokens (about one-tenth retention), LiteLVLM preserves over 10% higher performance than other methods, achieving 79.2% compared to 69.1% for VisionZip. These results suggest that strategies selecting globally informative tokens (*e.g.*, LLaVA-PruMerge, VisionZip) fail to retain tokens relevant to the referent regions, as token importance varies with the referring expression. Paradoxically, TRIM—which selects tokens with high visual-text similarity—suffers a severe performance drop of 10-20% compared to other methods. In contrast, LiteLVLM demonstrates its effectiveness by retaining text-aware visual tokens with low visual-text similarity and recovering context-aware tokens, substantially maintaining performance.

### 5.3. Referring Video Object Segmentation

**Implementation Details.** To extend our experiments to the video domain, we employ VideoGLaMM (Munasinghe et al., 2025), an extension of GLaMM designed for video inputs. Following the experimental setup of VideoLISA (Bai et al., 2024) and VideoGLaMM, we report Region Jaccard $\mathcal{J}$, Boundary F-measure $\mathcal{F}$, and their mean $\mathcal{J}\&\mathcal{F}$.

**Main Results.** Pixel grounding in videos is further complicated by higher visual redundancy and temporal inconsistency. As shown in Table 2 and Table 3, we evaluate LiteLVLM on referring video object segmentation benchmarks (Khoreva et al., 2018; Seo et al., 2020), comparing it against the latest VisPruner. With 65.9% of the visual tokens reduced (down to 196 tokens), **LiteLVLM remarkably preserves** 99.5% **of the vanilla VideoGLaMM performance on both benchmarks.** Even when further reduced to 81 tokens (85.9% reduction), LiteLVLM degrades by only 4.9% on Ref-DAVIS-17 and 3.9% on Refer-YouTube-VOS, whereas VisPruner drops by 13.1% and 10.1%, respectively. These outcomes demonstrate the effectiveness and generality of our method across both image and video domains. Moreover, LiteLVLM proves its versatility across various model architectures, including GLaMM and VideoGLaMM.

### 5.4. Ablation Study

**Ablations for the Core Components.** In Table 4, we report an ablation study to verify the effectiveness of the two components of LiteLVLM. To this end, we evaluate performance on the RefCOCO-*val* subset across varying token budgets: 288 (↓ 50%), 144 (↓ 75%), 58 (↓ 90%), and 29 (↓ 95%). Notably, model #1, based solely on context-aware tokens performs worse than model #0 with randomly pruned visual tokens (Avg. 63.3 vs. 65.0). As discussed in Section 3.2, selecting tokens with high visual-text similarity (model #2) leads to significant performance degradation compared to retaining low-similarity tokens in model #3 (Avg. 52.7 vs. 66.1). While model #3 maintains competitive performance, its focus on the foreground often loses the background context necessary for precise segmentation. By integrating low

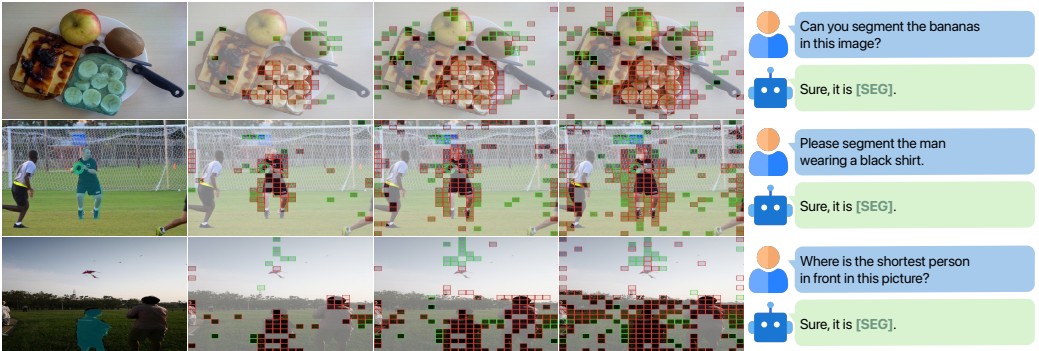

*Figure 5.* **Visualization of LiteLVLM for different referring expressions.** Similarity-aware and context-aware tokens are highlighted in red and green boxes, respectively. From left to right, the number of retained tokens is progressively increased (64, 128, and 192 tokens).

visual-text similarity visual tokens with recovered contextually informative ones, our LiteLVLM (model #4) yields a 2.7% gain in performance (Avg. 68.9%) and consistently achieves the best results across all token budgets.

**Ablations for the Token Selection Setup.** As described in Section 4.4, we design an adaptive token selection setup and analyze its impact in Table 5. To evaluate our design, we compare it against fixed ratios (25%, 50%, and 75%) of similarity-aware selection and context-aware recovery. Results show that while increasing the similarity-aware ratio from 25% to 50% improves the average performance ($67.6\% \rightarrow 68.9\%$), a further increase to 75% leads to a performance drop (68.1). This suggests that relying solely on similarity-aware tokens is suboptimal. In contrast, our adaptive setup consistently achieves the best performance across all budgets (Avg. 70.9%). This confirms that our approach adaptively determines the optimal number of referent-relevant tokens by leveraging the input prompts.

## 5.5. Efficiency Analysis

To verify the efficiency of LiteLVLM, we evaluate FLOPs, prefilling time, CUDA time, and cache storage compared with FastV and VisPruner on the RefCOCO-*val* subset. All metrics are measured end-to-end for inference on a single NVIDIA A100-40GB GPU. In Table 6, LiteLVLM reduces FLOPs by over $2\times$ and $3.5\times$ when retaining 192 and 64 tokens, respectively. Compared to the vanilla GLaMM, LiteLVLM achieves up to a 22% and 30.4% faster inference speed (CUDA time) while reducing activation memory (storing activation) by up to $3.9\times$. Since LiteLVLM prunes tokens before the LLM, it substantially accelerates first-token generation (prefilling time). This before-LLM pruning also enables compatibility with faster attention mechanisms such as FlashAttention (Dao et al., 2022), which is infeasible for FastV due to its reliance on in-LLM attention. Moreover, LiteLVLM exhibits higher efficiency than VisPruner, benefiting from its simpler computation for identifying important tokens. These results highlight LiteLVLM as a practical and lightweight solution for cost-efficient LVLM inference.

*Table 6.* **Efficiency analysis with LiteLVLM on RefCOCO-*val*.**

| Method | FLOPs (TB) | Prefilling Time (ms) | CUDA Time (ms) | Storing Activation (GB) |
|---|---|---|---|---|
| *Upper Bound, All 576 Tokens (100%)* | | | | |
| GLaMM | 4.66 | 166.25 | 340.89 | 0.81 |
| *Retain 192 Tokens (↓ 66.7%)* | | | | |
| FastV (ECCV24) | 2.65 | 162.88 | 340.25 | 0.81 |
| VisPruner (ICCV25) | 2.17 | 75.65 | 276.09 | 0.37 |
| **LiteLVLM** (ICML26) | **2.11** | **74.88** | **265.83** | **0.35** |
| *Retain 64 Tokens (↓ 88.9%)* | | | | |
| FastV (ECCV24) | 2.23 | 157.50 | 338.65 | 0.80 |
| VisPruner (ICCV25) | 1.33 | 54.77 | 253.10 | 0.23 |
| **LiteLVLM** (ICML26) | **1.27** | **54.02** | **237.35** | **0.21** |

## 5.6. Qualitative Visualization

Figure 5 presents qualitative results of LiteLVLM on RefCOCO-*val* subset across various referring expressions, with 64, 128, and 192 tokens retained. By leveraging textual information, similarity-aware tokens (red boxes) primarily focus on the referent regions, even in examples requiring relative or comparative reasoning (*e.g.*, "shortest person"). Meanwhile, context-aware tokens (green boxes) capture the global context, further facilitating accurate segmentation.

## 6. Conclusion

In this paper, we analyze CLIP and reveal an inversion in similarity between visual and text tokens. To tackle this, we introduce LiteLVLM, a simple yet effective training-free token pruning method for pixel grounding tasks. Our method retains visual tokens with low visual-text similarity while recovering contextually informative ones. LiteLVLM substantially reduces the number of visual tokens while preserving the model performance and improving computational efficiency. Extensive experimental results demonstrated that, with a 66.7% reduction in visual tokens, LiteLVLM achieves a 54.7% reduction in memory consumption and a 22% inference speedup, while maintaining 90.3% of the original performance. Furthermore, LiteLVLM extends effectively to the video modality, maintaining 99.5% of the original performance. LiteLVLM can facilitate the practical deployment of LVLMs on edge devices and resource-constrained servers by providing a cost-effective solution that reduces overhead without compromising grounding performance.

## Acknowledgements

This research was partly supported by the MSIT(Ministry of Science and ICT), Korea, under the ITRC(Information Technology Research Center) support program(IITP-2026-RS-2024-00437494) supervised by the IITP(Institute for Information & Communications Technology Planning & Evaluation); in part by the IITP(Institute of Information & Coummunications Technology Planning & Evaluation)-ICAN(ICT Challenge and Advanced Network of HRD) grant funded by the Korea government(Ministry of Science and ICT)(IITP-2026-RS-2022-00156345); in part by Unmanned Vehicles Core Technology Research and Development Program through the National Research Foundation of Korea (NRF) and Unmanned Vehicle Advanced Research Center (UVARC) funded by the Ministry of Science and ICT, the Republic of Korea (NRF-2023M3C1C1A01098408).

## Impact Statement

LiteLVLM enables the efficient deployment of off-the-shelf large vision-language models (LVLMs) in computationally constrained environments, such as edge devices and resource-limited servers or cloud platforms. As this work is primarily concerned with model inference efficiency and does not involve additional data collection, we believe the potential for misuse to be limited and that discussion of societal implications is not necessary in the current context.

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

# Appendix

## A. Dataset

We conduct experiments with LiteLVLM on 6 widely used benchmarks, including 3 referring expression segmentation datasets and 3 referring video object segmentation datasets. Each dataset is described in detail below.

### A.1. Referring Expression Segmentation

For fair comparison, we evaluate our method on 3 referring expression benchmarks following the experimental settings and evaluation protocols of GLaMM (Rasheed et al., 2024). All three datasets—RefCOCO, RefCOCO+, RefCOCOg—are built on MS COCO (Lin et al., 2014) images and segmentation masks.

**RefCOCO** (Kazemzadeh et al., 2014). RefCOCO is designed for the referring expression comprehension task and is used to evaluate pixel grounding performance. It provides multiple natural language expressions for each target object, together with bounding box and segmentation mask annotations. The dataset is split into validation (val), testA, and testB subsets, where testA primarily consists of expressions referring to person instances, while testB mainly includes expressions referring to non-person object categories.

**RefCOCO+** (Kazemzadeh et al., 2014). RefCOCO+ follows the setup of RefCOCO but differs in that the referring expressions do not contain explicit spatial words, such as "left/right" and "in front of/behind". Consequently, models are required to rely more on visual appearance and contextual reasoning, making the task more challenging. As in RefCOCO, the dataset is also split into validation (val), testA, and testB subsets.

**RefCOCOg** (Mao et al., 2016). RefCOCOg is annotated in a non-interactive, crowd-sourced process on Amazon Mechanical Turk (Buhrmester et al., 2016), generating longer and more complex referring expressions (on average 8-9 words, compared to 3-4 words in RefCOCO). These expressions are typically free-form and descriptive language, capturing object attributes (*e.g.*, color and material) and relative and comparative relationships. Unlike RefCOCO, which is split into val, testA, and testB to separate person and non-person expressions, RefCOCOg is split only into val and test subsets.

### A.2. Referring Video Object Segmentation

To extend LiteLVLM to the video domain and evaluate its performance, we additionally conduct experiments on 3 referring video object segmentation benchmarks used in VideoGLaMM (Munasinghe et al., 2025).

**Ref-DAVIS-17** (Khoreva et al., 2018). Ref-DAVIS-17 is a video object segmentation benchmark that extends a subset of DAVIS 2017 (90 sequences) with natural language expressions. Each video contains multiple object instances, where each object is annotated with one or more textual descriptions, enabling language-conditioned segmentation without first-frame mask initialization.

**Refer-YouTube-VOS** (Seo et al., 2020). Refer-YouTube-VOS is a large-scale referring video object segmentation benchmark constructed on the YouTube-VOS-2019 dataset, where each object instance is annotated with natural language expressions. It contains approximately 4k high-resolution videos with 27k referring expressions and dense pixel-level masks across frames, enabling evaluation of sustained instance tracking and segmentation.

**MeViS** (Ding et al., 2023). MeVis is a large-scale video segmentation benchmark designed for motion expression guided object segmentation, where objects cannot be identified from a single frame alone. It consists of 2k videos with dense pixel-level annotations across frames, featuring complex scenes with multiple moving objects that require temporal reasoning.

*Table 7.* **Performance comparison on MeViS.**

| Model | $\mathcal{J}$ | $\mathcal{F}$ | $\mathcal{J}\&\mathcal{F}$ | Rel. |
|---|---|---|---|---|
| *Upper Bound, All 576 Tokens* **(100%)** | | | | |
| VideoGLaMM | 42.07 | 48.23 | 45.15 | 100% |
| *Retain 196 Tokens* (↓ **65.9%**) | | | | |
| VisPruner | 41.93 | 47.96 | 44.94 | 99.5% |
| **LiteLVLM** | **42.06** | **48.10** | **45.08** | **99.8%** |
| *Retain 81 Tokens* (↓ **85.9%**) | | | | |
| VisPruner | 41.35 | 47.83 | 44.59 | 98.7% |
| **LiteLVLM** | **41.64** | **47.82** | **44.73** | **99.0%** |

*Table 8.* **Performance comparison of various pruning methods on LLaVA-1.5-7B.** Here, **Avg.** denotes the average accuracy across 7 LLaVA benchmarks, and **Rel.** represents the average percentage of performance maintained at the corresponding reduction ratio. LiteLVLM[‡] indicates that only 5% of similarity-aware tokens are retained, and the remaining tokens are selected from context-aware ones.

| Method | VQA$^{V2}$ | GQA | VQA$^{Text}$ | POPE | MMB | MMB$^{CN}$ | MMVet | Avg. | Rel. |
|---|---|---|---|---|---|---|---|---|---|
| *Upper Bound, All 576 Tokens (100%)* | | | | | | | | | |
| LLaVA-1.5-7B | 78.5 | 62.0 | 58.2 | 85.9 | 64.3 | 58.3 | 31.1 | 62.61 | 100.0% |
| *Retain 128 Tokens (↓ 77.8%)* | | | | | | | | | |
| ToMe (ICLR23) | 63.0 | 52.4 | 49.1 | 62.8 | 53.3 | 48.8 | 27.2 | 50.94 | 81.4% |
| FastV (ECCV24) | 61.8 | 49.6 | 50.6 | 59.6 | 56.1 | 51.4 | 28.1 | 51.03 | 81.5% |
| SparseVLM (ICML25) | 73.8 | 56.0 | 54.9 | 80.5 | 60.0 | 51.1 | 30.0 | 58.04 | 92.8% |
| LLaVA-PruMerge+ (ICCV25) | 74.7 | 57.8 | 54.3 | 81.5 | 61.3 | 54.7 | 28.7 | 59.00 | 94.2% |
| VisionZip (CVPR25) | 75.6 | 57.6 | 56.8 | 83.2 | 62.0 | 56.7 | 32.6 | 60.64 | 96.9% |
| VisPruner (ICCV25) | 75.8 | 58.2 | **57.0** | 84.6 | **62.7** | **57.3** | **33.7** | 61.32 | **97.9%** |
| **LiteLVLM** (ICML26) | 76.1 | 58.1 | 56.9 | 84.2 | 61.6 | 56.1 | 31.7 | 60.69 | 96.9% |
| **LiteLVLM**[‡] (ICML26) | **76.6** | **58.4** | **57.0** | 85.3 | 62.5 | 56.6 | 32.9 | **61.33** | **97.9%** |
| *Retain 64 Tokens (↓ 88.9%)* | | | | | | | | | |
| ToMe (ICLR23) | 57.1 | 48.6 | 45.3 | 52.5 | 43.7 | 38.9 | 24.1 | 44.31 | 70.8% |
| FastV (ECCV24) | 55.0 | 46.1 | 47.8 | 48.0 | 48.0 | 42.7 | 25.8 | 44.77 | 71.5% |
| SparseVLM (ICML25) | 68.2 | 52.7 | 51.8 | 75.1 | 56.2 | 46.1 | 23.3 | 53.34 | 85.2% |
| LLaVA-PruMerge+ (ICCV25) | 67.4 | 54.9 | 53.0 | 77.4 | 59.3 | 51.0 | 25.9 | 55.56 | 88.8% |
| VisionZip (CVPR25) | 72.4 | 55.1 | 55.5 | 77.0 | 60.1 | **55.4** | 31.7 | 58.17 | 92.9% |
| VisPruner (ICCV25) | 72.7 | 55.4 | **55.8** | **80.4** | 61.3 | 55.1 | **32.3** | **59.00** | **94.2%** |
| **LiteLVLM** (ICML26) | 72.8 | 55.5 | 53.5 | 78.2 | 60.1 | 54.0 | 29.6 | 57.96 | 92.6% |
| **LiteLVLM**[‡] (ICML26) | **73.2** | **56.1** | 53.6 | 80.1 | **61.4** | 55.1 | 30.4 | 58.56 | 93.5% |
| *Retain 32 Tokens (↓ 94.4%)* | | | | | | | | | |
| ToMe (ICLR23) | 46.8 | 43.6 | 38.3 | 39.0 | 31.6 | 28.1 | 17.3 | 34.96 | 55.9% |
| FastV (ECCV24) | 43.4 | 41.5 | 42.6 | 32.5 | 37.8 | 33.2 | 20.7 | 36.02 | 57.5% |
| SparseVLM (ICML25) | 58.6 | 48.3 | 57.3 | 67.9 | 51.4 | 40.6 | 18.6 | 48.96 | 78.2% |
| LLaVA-PruMerge+ (ICCV25) | 54.9 | 51.1 | 50.6 | 70.9 | 56.8 | 47.0 | 21.4 | 50.39 | 80.5% |
| VisionZip (CVPR25) | 67.1 | 51.8 | 53.1 | 68.7 | 57.7 | 50.3 | 25.5 | 53.03 | 84.7% |
| VisPruner (ICCV25) | 67.7 | 52.2 | **53.9** | 72.7 | **58.4** | **52.7** | 28.8 | 55.35 | 88.4% |
| **LiteLVLM** (ICML26) | 67.2 | 52.5 | 52.9 | 71.5 | 56.9 | 49.3 | 24.9 | 53.60 | 85.7% |
| **LiteLVLM**[‡] (ICML26) | **67.8** | **52.8** | 53.8 | **74.5** | 57.7 | 51.9 | **29.1** | **55.37** | **88.5%** |

# B. Additional Quantitative Results

### B.1. Referring Video Object Segmentation

We present additional quantitative evaluations of LiteLVLM on the MeViS dataset (Ding et al., 2023) for referring video object segmentation. As described in Section 5.3 of the main paper, we apply LiteLVLM within VideoGLaMM (Munasinghe et al., 2025) to ground instances specified via referring expressions across video frames. As shown in Table 7, LiteLVLM maintains its performance with only a 0.2% drop while pruning 65.9% of the total visual tokens (192 tokens). Even when pruning 85.9% of the visual tokens (81 tokens), LiteLVLM still preserves 99.0% of the original performance. Moreover, LiteLVLM consistently outperforms the state-of-the-art method, VisPruner (Zhang et al., 2025a), validating the effectiveness of our method (VisPruner vs. LiteLVLM in **Rel.**: 99.5% vs. 99.8% with 192 tokens, and 98.7% vs. 99.0% with 81 tokens).

### B.2. Image Understanding Tasks

Although our method is fundamentally designed for efficient pixel-level grounding, we evaluate its performance on image understanding tasks to examine how it compares to other token pruning methods commonly studied in prior work. To this end, we conduct experiments on 7 widely used image understanding multi-modal benchmarks, including VQAv2 (Goyal et al., 2017), GQA (Hudson & Manning, 2019), VQA$^{Text}$ (TextVQA) (Singh et al., 2019), POPE (Li et al., 2023b), MMBench (MMB) (Liu et al., 2024d), MMBench-CN (MMB$^{CN}$) (Liu et al., 2024d), and MMVet (Yu et al., 2024). We apply LiteLVLM to the classic LLaVA-1.5 (Liu et al., 2024a) with Vicuna-7B (Zheng et al., 2023) model and compare it with previous state-of-the-art methods: ToMe (Bolya et al., 2023), FastV (Chen et al., 2024a), SparseVLM (Zhang et al., 2025b), LLaVA-PruMerge+ (Shang et al., 2025), VisionZip (Yang et al., 2025), and VisPruner (Zhang et al., 2025a). For a fair comparison,

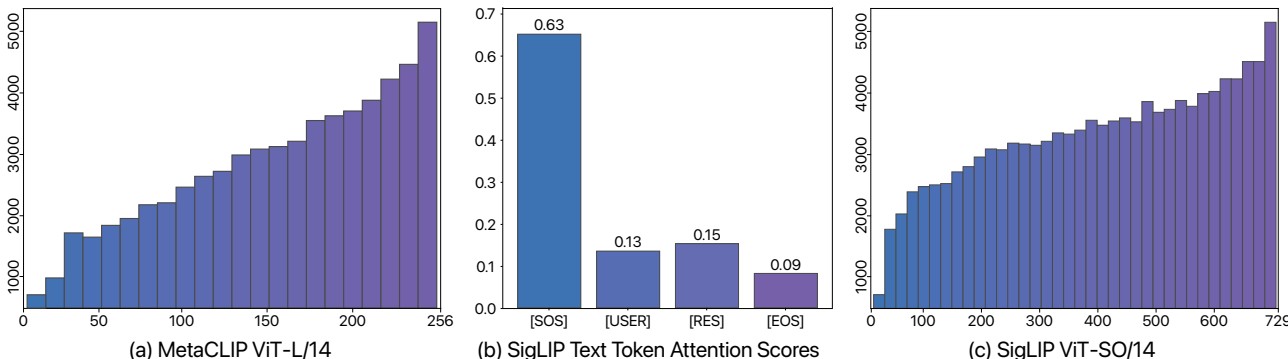

*Figure 6.* **Deeper analysis of CLIP-family variant encoders.** (a) MetaCLIP [REF]-[EOS] similarity rank distribution. (b) SigLIP average attention scores from the [EOS] to each text token. (c) SigLIP [REF]-[EOS] similarity rank distribution.

all evaluations under the settings reported in the original LLaVA paper. Table 8 presents the experimental results when retaining 128, 64, and 32 visual tokens. When the number of tokens is reduced from 576 to 128 (↓ 77.8%), LiteLVLM maintains competitive performance across LLaVA benchmarks, achieving an average accuracy (**Avg.**) of 60.69% with 96.9% performance retention (**Rel.**). When the number of tokens is further reduced to 64 (↓ 88.9%) and 32 (↓ 94.4%), LiteLVLM preserves 92.6% and 85.7% of the original performance, respectively. Overall, LiteLVLM demonstrates better performance than most prior methods (*e.g.*, SparseVLM and LLaVA-PruMerge+) and achieves performance comparable to the latest VisPruner. In particular, LiteLVLM outperforms text-guided methods such as FastV and SparseVLM, highlighting the effectiveness of selecting low visual-text similarity tokens. Additionally, we evaluate LiteLVLM‡, which retains only 5% text-guided similarity-aware tokens while selecting the majority of tokens as context-aware, considering the nature of image understanding tasks. Interestingly, LiteLVLM‡ consistently outperforms LiteLVLM and achieves performance comparable to, or even surpassing, the state-of-the-art VisPruner with 128 and 32 tokens. From these results, we argue that LiteLVLM effectively retains tokens relevant to natural language while recovering contextually important visual information, and that controlling the number of context-aware tokens further supports strong generalization in image understanding tasks.

## C. Extended CLIP-family Analysis

In Section 3.2, we reveal a visual-text similarity reversal in CLIP, where visual tokens belonging to referent regions tend to show low similarity to the global text representation. To examine whether this phenomenon persists across more recent fine-grained visual encoders, we further extend our analysis to two CLIP-family variants: MetaCLIP (Chuang et al., 2026) and SigLIP (Zhai et al., 2023). MetaCLIP builds upon CLIP by scaling up the pretraining data and improving data quality. In contrast, SigLIP replaces the contrastive learning objective with a sigmoid-based loss. As illustrated in Figure 6-(a), MetaCLIP still holds the visual-text similarity reversal. Specifically, MetaCLIP ViT-L/14 encodes a resized 224 × 224 image with a patch size of 14, producing 256 visual tokens, where the [REF] tokens are consistently rank low. In Figure 6-(b), SigLIP retains a similar attention bias as observed in Figure 3-(a), where the [EOS] token remains heavily biased toward the [SOS] token, suggesting that the text attention sink artifact persists even under the sigmoid-based training objective. Consequently, Figure 6-(c) shows that the [REF]-[EOS] similarity ranks in SigLIP ViT-SO/14 are likewise concentrated near 729, indicating that the visual-text similarity reversal generalizes beyond OpenAI CLIP to diverse CLIP-family variants.

## D. Generalization Experiments

To validate the generalization of LiteLVLM beyond LLaVA and CLIP, we further extend our experiments to a different LVLM and vision encoder. We first evaluate our method on Qwen2.5-VL (Bai et al., 2025b) to assess its compatibility with a different LVLM architecture. Then, we perform experiments using SigLIP-2 (Tschannen et al., 2025) as the vision encoder.

### D.1. Generalization Beyond LLaVA

Since LiteLVLM is primarily studied on LLaVA-1.5, we additionally test our method on UniPixel (Liu et al., 2026), which is built upon Qwen2.5-VL. Concretely, Qwen2.5-VL employs a redesigned Vision Transformer (ViT) as its vision encoder to support native input resolutions, while adopting Qwen2.5 LLM. In Table 9, we compare LiteLVLM with recent LLaVA-PruMerge and VisionZip on the RefCOCO/+/g-*val* subsets with UniPixel, where the upper bound consists of

*Table 9.* **Performance comparison of token pruning methods on Qwen2.5-VL-based LVLM.**

| Method | RefCOCO | RefCOCO+ | RefCOCOg | Avg. |
|---|---|---|---|---|
| *Upper Bound, All 345 Tokens* **(100%)** | | | | |
| UniPixel (NeurIPS25) | 80.5 | 74.3 | 76.3 | 77.0 |
| *Retain 114 Tokens* (↓ **66.7**%) | | | | |
| LLaVA-PruMerge (ICCV25) | 75.2 | 65.8 | 70.8 | 70.6 |
| VisionZip (CVPR25) | 77.2 | 68.0 | 71.7 | 72.3 |
| **LiteLVLM** (ICML26) | **79.2** | **68.3** | **75.6** | **74.4** |
| *Retain 78 Tokens* (↓ **88.9**%) | | | | |
| LLaVA-PruMerge (ICCV25) | 72.1 | 60.9 | 66.6 | 66.5 |
| VisionZip (CVPR25) | 73.0 | **62.1** | 67.5 | 67.5 |
| **LiteLVLM** (ICML26) | **74.1** | **62.1** | **70.1** | **68.8** |

*Table 10.* **Performance comparison of token pruning methods on SigLIP-2-based LVLM.**

| Method | RefCOCO | RefCOCO+ | RefCOCOg | Avg. |
|---|---|---|---|---|
| *Upper Bound, All 729 Tokens* **(100%)** | | | | |
| X-SAM (AAAI26) | 85.1 | 78.0 | 83.8 | 82.3 |
| *Retain 244 Tokens* (↓ **66.7**%) | | | | |
| LLaVA-PruMerge (ICCV25) | 69.6 | 55.0 | 62.9 | 62.5 |
| VisionZip (CVPR25) | 78.8 | 67.5 | 75.3 | 73.9 |
| **LiteLVLM** (ICML26) | **79.4** | **70.8** | **76.9** | **75.7** |
| *Retain 162 Tokens* (↓ **77.8**%) | | | | |
| LLaVA-PruMerge (ICCV25) | 70.1 | 58.5 | 60.1 | 62.9 |
| VisionZip (CVPR25) | 75.9 | 62.4 | 72.1 | 70.1 |
| **LiteLVLM** (ICML26) | **76.6** | **64.0** | **72.9** | **72.2** |
| *Retain 81 Tokens* (↓ **88.9**%) | | | | |
| LLaVA-PruMerge (ICCV25) | 61.8 | 50.2 | 59.5 | 57.2 |
| VisionZip (CVPR25) | 69.5 | 51.6 | 63.8 | 61.6 |
| **LiteLVLM** (ICML26) | **69.6** | **54.8** | **65.9** | **63.4** |

345 visual tokens. We evaluate all methods under pruned token budgets of 114 (↓66.7%) and 78 (↓88.9) retained tokens. Across all token budgets, LiteLVLM consistently achieves the best performance, outperforming existing token pruning methods. Notably, LiteLVLM retains 96.7% of the original performance with 66.7% token pruning (77.0 → 74.4 Avg.), while still preserving 89.4% performance under 88.9% token pruning (77.0 → 68.8 Avg.). From these results, we argue that LiteLVLM is not tightly coupled to LLaVA and effectively generalizes across different LVLM architectures.

### D.2. Generalization Beyond CLIP

We next assess the versatility of LiteLVLM beyond the CLIP-based vision encoder. To this end, we conduct experiments on X-SAM (Wang et al., 2026), which is a state-of-the-art LVLM built upon the SigLIP-2 (Tschannen et al., 2025) vision encoder. Compared to CLIP, SigLIP-2 introduces additional localization and dense prediction losses (LocCa and SILC/TIPS), making token pruning methods based on global visual representations more challenging. Notably, since SigLIP-2 does not employ dedicated [CLS] and [EOS] tokens, we treat the visual token receiving average attention from all other tokens as the [CLS] token and the mean embedding of all text tokens as the [EOS] representation. As shown in Table 10, LiteLVLM demonstrates superior performance over previous token pruning methods (LLaVA-PruMerge and VisionZip) across all token budgets. LiteLVLM preserves 97.6% of the full 729 token performance under 66.7% token pruning and still maintains 91.7% performance even when 88.9% of the visual tokens are removed. Taken together, these results suggest that our LiteLVLM is broadly applicable across diverse vision encoders without being bound to CLIP-specific representations.

## E. Extended Ablation Study

### E.1. Impact of Fine-Tuning

Although LiteLVLM demonstrates strong performance without training or fine-tuning, we additionally evaluate the effect of fine-tuning on its performance. In Table 11, we compare the RefCOCO-*val* performance of LiteLVLM on GLaMM with and

*Table 11.* **Impact of fine-tuning LiteLVLM on RefCOCO-*val*.** FT denotes the fine-tuning, while models without FT are evaluated in a zero-shot manner. Here, **Rel.** indicates the average percentage of performance maintained at each reduction ratio.

| Method | FT | cIoU | Rel. |
|---|---|---|---|
| GLaMM | ✓ | 79.5 | 100% |
| LiteLVLM | ✗ | 74.4 | 93.6% |
| w/ Retain 192 Tokens (↓ 66.7%) | ✓ | **77.4** | **97.3%** |
| LiteLVLM | ✗ | 72.1 | 90.7% |
| w/ Retain 128 Tokens (↓ 77.8%) | ✓ | **76.9** | **96.7%** |
| LiteLVLM | ✗ | 66.3 | 83.4% |
| w/ Retain 64 Tokens (↓ 88.9%) | ✓ | **74.3** | **93.4%** |

*Table 12.* **Memory and performance of LiteLVLM on LLaVA-1.5-7B with the quantization.** Here, **Avg.** denotes the average accuracy across 4 LLaVA benchmarks, and **Rel.** indicates the average percentage of performance retention. Memory refers to the practical CUDA memory usage on a single NVIDIA A100-40GB GPU for SQA benchmark.

| Method | Memory (MB) | VQA$^{V2}$ | GQA | POPE | SQA | Avg. | Rel. |
|---|---|---|---|---|---|---|---|
| *Upper Bound, All 576 Tokens* **(100%)** | | | | | | | |
| LLaVA-1.5-7B | 16022.16 | 78.5 | 62.0 | 85.9 | 70.7 | 74.28 | 100% |
| *Retain 128 Tokens* (↓ **77.8%**) | | | | | | | |
| LiteLVLM | 15903.98 | 76.6 | 58.4 | **85.3** | **70.3** | **72.65** | **97.8%** |
| LiteLVLM-8bit | 9606.90 | **76.8** | **58.5** | 84.8 | 69.8 | 72.48 | 97.6% |
| LiteLVLM-4bit | **6504.32** | 76.5 | 58.1 | 84.7 | 69.4 | 72.18 | 97.2% |
| *Retain 32 Tokens* (↓ **94.4%**) | | | | | | | |
| LiteLVLM | 15902.65 | 67.8 | **52.8** | **74.5** | **70.1** | **66.30** | **89.3%** |
| LiteLVLM-8bit | 9497.26 | **67.9** | **52.8** | 74.2 | 69.6 | 66.13 | 89.0% |
| LiteLVLM-4bit | **6409.60** | 67.7 | 52.5 | 74.1 | 68.8 | 65.78 | 88.6% |

without fine-tuning (**FT**). At a 66.7% reduction ratio, fine-tuning the model with LiteLVLM improves performance by 3%. The performance gap widens as the token count is further reduced, yielding gains of respectively 4.8% and 8% at 77.8% and 88.9% reduction ratios. These results indicate that LiteLVLM still maintains reasonable performance even without training, while allowing clear performance improvements from fine-tuning when data is available.

### E.2. More Efficiency Analysis

To reduce memory usage, we conduct experiments with quantization techniques on LiteLVLM. As shown in Table 12, we evaluate LLaVA-1.5-7B performance and CUDA memory consumption of LiteLVLM with 8-bit and 4-bit quantization on the SQA benchmark (Lu et al., 2022) following VisionZip (Yang et al., 2025). Here, we employ LiteLVLM$^{\ddagger}$, an enhanced version that primarily uses context-aware tokens. When retaining 192 and 32 visual tokens, LiteLVLM alone yields only a marginal memory reduction (within 1%, ∼118-120 MB). However, quantizing LiteLVLM further reduces memory usage substantially—by about 40% with 8-bit quantization (16022.16 → 9606.90 MB) and 59% with 4-bit quantization (16022.16 → 6504.32 MB)—while preserving performance. These results demonstrate that quantized LiteLVLM achieves significantly lower CUDA memory usage without sacrificing performance, leading to improved efficiency.

## F. Additional Qualitative Results

### F.1. Visualization of Visual-Text Similarity Reversal

In Figure 7, we visualize the highest (green boxes) and lowest (red boxes) 10 visual tokens selected based on CLIP visual–text similarity for the RefCOCO+ (top row) and RefCOCOg (bottom row) benchmarks, where the left image shows the segmentation and the right image illustrates the selected tokens. Overall, low-similarity visual tokens are densely concentrated on the referent, whereas high-similarity tokens appear scattered, largely focusing on background regions. As discussed in Section 3.2 of the main paper, this observation illustrates the visual-text similarity reversal, showing that CLIP visual tokens with lower similarity to the text token, *i.e.*, the [EOS] token, effectively encode object information, whereas high-similarity tokens tend to focus on globally informative regions (*e.g*, background).

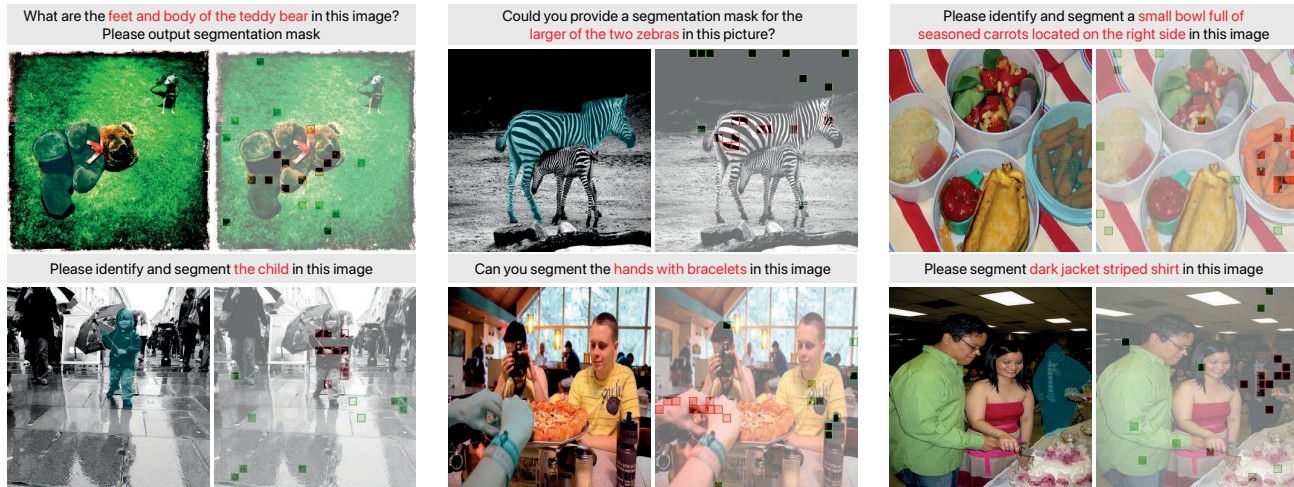

*Figure 7.* **Visualization of CLIP Visual-Text Similarity Reversal.** The top row shows RefCOCO+ results, while the bottom row presents RefCOCOg benchmark results, with green- and red-outlined tokens indicating the highest and lowest visual–text similarity, respectively.

## F.2. More Visualization Examples

Figure 8 depicts qualitative examples of LiteLVLM on two referring video object segmentation benchmarks (Seo et al., 2020; Ding et al., 2023). The top three rows show examples from the Refer-YouTube-VOS dataset, and the bottom three from the MeViS dataset. For clearer and more interpretable visualizations, we display only 81 similarity-aware visual tokens and mark the segmentation target (*e.g.*, black bear, white sedan) with red text, while also highlighting the corresponding segmented region in red. These visualizations demonstrate that LiteLVLM adaptively retains visual tokens corresponding to the referent that moves frame to frame.

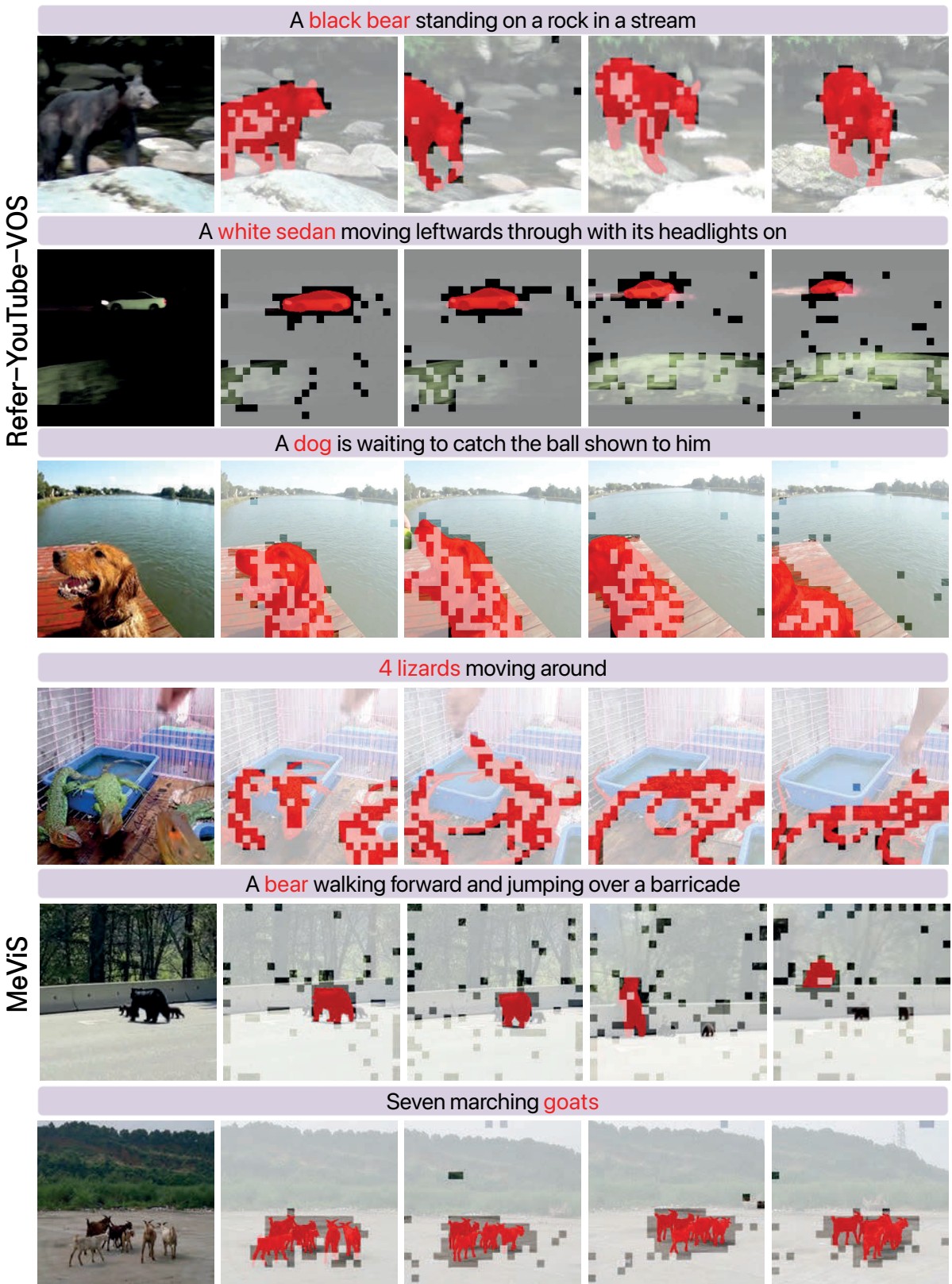

*Figure 8.* **More qualitative visualizations of LiteLVLM on referring video object segmentation.**

