# OpenReview forum: "CLIP Tricks You: Training-free Token Pruning for Efficient Pixel Grounding in Large Vision-Language Models"
_ICML.cc/2026/Conference — ICML 2026 regular_

### Official Review · Reviewer_rYHr · 2026-03-03

**Soundness:** 3
**Presentation:** 3
**Significance:** 2
**Originality:** 2
**Overall Recommendation:** 4
**Confidence:** 4

**Summary:**

The paper propose an interesting, simple and training-free method to prune visual tokens in vlms for the pixel grounding task.
The idea is based on the attention sink in the clip text encoder, where the eos token mainly attends on the sos token rather than other tokens.
Due the contrastive mechanism of aligning text eos token and imagae cls token, thus the image cls token also does not well attend to the other visual tokens of the same image.
The pixel grounding task needs the fine-grained visual info rather than the rough global info, so the paper picks the visual tokens with the low similarity with the text eos token.

**Compliance With Llm Reviewing Policy:**

Affirmed.

**Final Justification:**

The rebuttals address my concerns.

**Key Questions For Authors:**

All above bullets are my concerns. I would increase the score from weak reject to weak accept if other reviewers are happy about the method under the main limitation that the method would be ineffective when attention sink does not exit.

**Limitations:**

The paper does not discuss the limitations. The main limitation is that when the attention sink does not happen in the latest models, the method could be limited.

**Strengths And Weaknesses:**

First, the experiments are solid, specifically the tables 1 and 8 show the effectiveness of this prunning method, holding the performances for both the understanding task and pixel grounding tasks.

Second, my main concerns include two points:

1) the paper heavily steps on the clip, which has an obvious attention sink problem in the model. It would be better to show the other models, like SigLIP, which is also trained using contrastive way but sightly different loss. If it is also true, it would enhance the generalization of the method.

2) the method is based on the attention sink problem, which brings more and more attention from the research community to this problem. In the modern language models, the training already starts to apply some techniques to avoid the attention sink. If the new clip-like models or the next generation of vision encoders do not have the attention sink, the method will be limited and unlikely used, it this true?

---

In figure 2, a type in the title of the third subfigure, the title should be [REF] - [EOS].
For the figure 4, it would be better to make the arrows bigger.

---

> ### Author Rebuttal · Authors · 2026-03-31
>
> We sincerely thank the reviewer `rYHr` for the effort devoted to reviewing our paper and for acknowledging the `solid` experimental results demonstrating the effectiveness of our method. Our responses to the reviewer's comments are summarized as follows.
>
> ---
>
> > **W1: Attention Sink Beyond CLIP.**
>
> We understand the reviewer's concern that the attention sink phenomenon is analyzed only in CLIP, and that validating it across other models would enhance the generalization of our method. To this end, we investigate SigLIP, which is pre-trained with a sigmoid-based loss, and study both the text attention sink and visual-text similarity reversal. As illustrated in Figure [(link)](https://anonymous.4open.science/r/CLIP-Tricks-You/rebuttal/analysis_SigLIP.png), we observe that SigLIP shows similar tendencies as CLIP (Figures 2 and 3 in the main paper), where [EOS] strongly attends to [SOS] and [REF] tokens tend to rank low in similarity to [EOS]. That is, our observed phenomena are not specific to CLIP, and we believe they fundamentally arise from the global image-text alignment objective.
>
> ---
>
> > **W2 & L: Discussion: LiteLVLM Beyond Attention Sink.**
>
> As mentioned, we are aware of recent techniques [1, 2] to avoid attention sink in modern language models (LLMs). However, we would like to clarify that, as discussed in `W1` above, visual-text similarity reversal is largely driven by the global image-text alignment objective, while attention sink serves to accelerate this phenomenon. Therefore, even if next-generation CLIP-family models or other vision encoders mitigate attention sink through training techniques (*e.g.,* gated attention or softmax variants), it remains unclear as of now whether this similarity bias would be resolved. That said, we fully agree that investigating widely used CLIP-family encoders (*e.g.,* CLIP, MetaCLIP) under attention sink-free settings would be of interest for future work. In the current context, we would like to highlight that our complementary design, which first selects similarity-aware [REF] tokens and then recovers context-aware tokens, extends token pruning beyond conventional image-level understanding to object-centric tasks, providing a practical basis for which tokens to retain.
>
> [1] *Qiu et al., Gated Attention for Large Language Models: Non-linearity, Sparsity, and Attention-Sink-Free, NeurIPS, 2025.*
>
> [2] *Zuhri et al., Softpick: No Attention Sink, No Massive Activations with Rectified Softmax, arXiv:2504.20966, 2025.*
>
> ---
>
> > **W3: Figure Fixes.**
>
> We thank the reviewer for pointing out these issues we missed. We will correct the typo in Figure 2 [(link)](https://anonymous.4open.science/r/CLIP-Tricks-You/rebuttal/figure2_rebuttal.png) and improve the visibility in Figure 4 [(link)](https://anonymous.4open.science/r/CLIP-Tricks-You/rebuttal/figure4_rebuttal.png) in the final version.
>
> ---
>
> We sincerely appreciate your valuable review and suggestions. If you find our responses satisfactory, we would be grateful for your reconsideration of the rating.

---

> > ### Author Rebuttal · Reviewer_rYHr · 2026-03-31
> >
> > My questions are fully solved and I also have checked the other reviewers' comments and responses for them. I would love to improve the rating to WA.

---

> > > ### Author Response · Authors · 2026-04-01
> > >
> > > Dear **Reviewer rYHr**,
> > >
> > > We sincerely thank you for your valuable feedback and for recognizing our efforts in the revised analysis and discussion. We believe studying token pruning under attention sink-free settings is an important future direction and remains an open problem.
> > >
> > > We are grateful for your updated rating. We are happy to answer any further questions.

---

### Official Review · Reviewer_F6VR · 2026-03-12

**Soundness:** 3
**Presentation:** 3
**Significance:** 3
**Originality:** 3
**Overall Recommendation:** 4
**Confidence:** 4

**Summary:**

This paper proposes LiteLVLM, a training-free visual token pruning method for efficient pixel-level grounding in LVLMs. By identifying and leveraging the visual-text similarity reversal phenomenon in CLIP, the method retains low-similarity visual tokens while recovering context-aware tokens based on [CLS] attention. The approach is simple, plug-and-play, and demonstrates strong empirical performance with reduced computational overhead.

**Compliance With Llm Reviewing Policy:**

Affirmed.

**Key Questions For Authors:**

Please see the “Weaknesses” section above.

**Limitations:**

While the proposed LiteLVLM method demonstrates strong empirical results, several limitations remain. First, the pruning strategy relies on observations derived from CLIP’s internal behavior, which may limit generalization to other vision-language encoders. Second, the adaptive token selection introduces hyperparameters (e.g., the ratio of similarity-aware tokens) whose sensitivity is not fully analyzed. Finally, the evaluation mainly focuses on referring expression segmentation benchmarks, and it remains unclear how well the approach generalizes to other grounding or multimodal reasoning tasks.

**Strengths And Weaknesses:**

Strength:

1. The paper is well motivated. It clearly identifies the mismatch between visual token pruning strategies and text-conditioned grounding tasks, and provides a principled analysis of why existing similarity-based pruning methods may fail under pixel-level grounding settings.

2. The problem is carefully decomposed and analyzed. The investigation of visual-text similarity reversal and the text attention sink phenomenon makes the technical reasoning easy to follow and convincing.

3. The proposed method is simple, training-free, and practically effective.

 Weakness:

1. The notation throughout the paper is somewhat heavy and occasionally confusing. Some symbols are difficult to align with the figures. For example, the relationship between $Z_v$ and $E_j^V$ is not explicitly clarified. Similarly, the connections among $s_{i,j}$, $s_i'$, and the candidate set $S_i$ could be better explained.

2. In Section 4.4, the number of similarity-aware tokens is set to 50% of the budget. It is unclear how this hyperparameter is chosen. Is this empirically tuned? How sensitive is performance to this ratio?

3. While the paper focuses on pruning visual tokens based on the analysis of global representations in CLIP, it would be interesting to discuss whether a similar pruning strategy could be applied to text tokens. Could removing redundant text tokens further improve efficiency?

4. Since the analysis suggests that [EOS] and [CLS] tokens mainly capture global information, have the authors considered studying the effect of modifying or removing these tokens? An ablation analysis in this direction may provide additional insights into the proposed mechanism.

---

> ### Author Rebuttal · Authors · 2026-03-31
>
> We thank the reviewer `F6VR` for the `positive` comments and `high` confidence in assessing our work. We are glad that you found our observed phenomena clear and convincing. Below, we address each of your concerns in turn.
>
> ---
>
> > **W1: Improving Notation Clarity and Explanation.**
>
> We appreciate the opportunity to resolve confusion and clarify the notation. We will revise the notation to better align the symbols with Figure 4 in the final version.
>
> - **Relationship between $Z_v$ and $E_j^v$.**
>
> In the main paper, we denote $Z_v$ as the set of raw visual tokens ($Z_v=\\{v_j\\}_{j=1}^M$) and $E_j^v$ as the $j$-th projected visual token ($E_j^v=W \\cdot v_j$) used for similarity calculation. We will unify the notation to $z$ for both visual tokens and their projected embeddings. To ensure consistency, we have updated Figure 4 to reflect the unified symobls.
>
> - **Connections among $s_{i,j}, s_i^{'}$, and the candidate set $S_i$.**
>
> We first fill the candidate set $S_i$ for the $i$-th prompt by selecting the low-$k$ visual tokens based on similarity scores $s_{i,j}$ computed with $j$-th visual tokens. From the remaining unselected tokens, we recover top-scoring ones using context-aware scores $s_i^{'}$ to fill the budget ($B-|S_i|$). We will unify the visual token index to $j$ and update the context-aware scores accordingly ($s_i^{'}$ to $s_j^{'}$) to ensure strict correspondence.
>
> ---
>
> > **W2 & L2: Ablation of Token Selection Ratio.**
>
> While Table 5 in the main paper evaluates the impact of similarity-aware token ratios $(r \\in \\{0.25, 0.5, 0.75\\})$, we appreciate the suggestion for a more granular view. We expand the ablations to cover the full range of $r$ from 0.1 to 0.9 on RefCOCO-*val* subset with 192 retained tokens.
>
> **Table: Ablation of similarity-aware token ratio**
> |**Similarity-aware ($r$)**|**Context-aware**|**cIoU**|
> |:-|:-:|:-:|
> |**Retain 192 Tokens (&darr; 66.7%)**|||
> |0.1|0.9|70.7|
> |0.2|0.8|70.8|
> |0.3|0.7|71.5|
> |0.4|0.6|72.8|
> |0.5|0.5|**73.3**|
> |0.6|0.4|73.2|
> |0.7|0.3|72.8|
> |0.8|0.2|72.5|
> |0.9|0.1|72.2|
>
> - **Discussion:** Given that similarity-aware tokens focus on referent regions, the optimal number of tokens ideally varies with the object size. However, since this information is not available a priori, we empirically studied and show that LiteLVLM achieves peak performance when $r$ is tuned to 0.5. As $r \\to 0.1$, performance degrades due to limited referent coverage, while higher ratios (*e.g.,* $r \\to 0.9$) lose semantic and background context. While performance shows sensitivity to $r$, even at $r=0.1$, it remains within 2.6% of the peak.
>
> ---
>
> > **W3: Discussion on Text Token Pruning.**
>
> We thank the reviewer for valuable insight on pruning text tokens. Our work primarily focuses on pruning visual tokens, as visual tokens dominate the input sequence (hundreds to thousands), while text tokens are typically fewer than 100, leading to relatively modest efficiency improvements. Moreover, in tasks such as referring expression segmentation, while multiple visual tokens can redundantly cover the referent region, a few text tokens often uniquely encode the key semantics, and their removal may significantly degrade performance. However, we believe this is a promising direction for future research, particularly for longer textual inputs.
>
> ---
>
> > **W4: Effect of Modifying [CLS]/[EOS] Tokens.**
>
> The reviewer suggests analyzing the effect of modifying or removing [CLS] and [EOS] tokens to provide additional insights. We agree that this analysis can provide additional insight into the robustness of our mechanism, and we therefore conduct the following ablations. Following VisionZip, we compute the average attention each token receives from all others and use it to replace the [CLS] token (Avg. Attn.). We also replace the [EOS] token with the mean of all text tokens (Avg. Text), excluding the [SOS] and [EOS] tokens.
>
> **Table: Effect of replacing [CLS]/[EOS] tokens**
> |**Method**|**RefCOCO**|**RefCOCO+**|**RefCOCOg**|**Avg.**|
> |:-|:-|:-:|:-:|:-:|
> |**GLaMM**|79.5|72.6|74.2|75.4|
> |**Retain 192 Tokens (&darr; 66.7%)**|||||
> |LiteLVLM|74.4|64.1|66.0|**68.2**|
> |Avg. Attn.|74.2|63.7|66.0|68.0 (-0.2)|
> |Avg. Text|73.9|63.3|65.8|67.7 (-0.5)|
>
> - **Discussion:** As shown above, even after replacing the [CLS]/[EOS] tokens, the performance drops by less than 0.5%. These results suggest that our token selection mechanism is not tightly coupled to the original [CLS]/[EOS] representations.
>
> ---
>
> > **L1 & L3: Generalization Beyond CLIP and Other Tasks.**
>
> We recognize the reviewer's concern. Due to the rebuttal length limit, we kindly refer the reviewer to the visualization results in our response to Reviewer `FZx9 (W1-2 & Q1-2)` and the performance validation on the SigLIP-2-based model in our response to Reviewer `UJ5Y (Q1)`. Additionally, to assess generalization across different tasks, we evaluate LiteLVLM on 7 LLaVA benchmarks in Appendix Table 8, and encourage the reviewer to consult these results.

---

> > ### Author Rebuttal · Reviewer_F6VR · 2026-04-05
> >
> > Thank you for the detailed and constructive rebuttal. The clarifications on notation and the additional ablation on the token ratio help address my concerns about clarity and hyperparameter sensitivity. I also appreciate the additional experiments on newer LVLMs, which strengthen the empirical validation and partially address generalization concerns. The method is still largely motivated by properties of CLIP-style representations, and it is unclear how well it would generalize to substantially different vision encoders. In addition, while the ablation on the token ratio is useful, a more principled or adaptive strategy for selecting this ratio would further strengthen the work.

---

> > > ### Author Response · Authors · 2026-04-06
> > >
> > > We are glad that our responses and additional experiments improved the clarity of our work. We thank you for the thoughtful follow-up regarding generalization and the token selection ratio. Below, we address your specific concerns regarding generalization and the selection strategy.
> > >
> > > ---
> > >
> > > **1. Generalization Beyond CLIP Encoders.**
> > >
> > > We acknowledge that LiteLVLM is motivated by artifacts observed in CLIP. However, we respectfully note that CLIP and its variants (e.g., Meta-CLIP, SigLIP) remain widely used as vision encoders in recent LVLMs. More importantly, our method is not tied to CLIP-style representations but instead identifies which tokens are important for precise pixel grounding based on the analysis of visual-text similarity. If a new vision encoder shows high similarity between [REF] tokens and [EOS] embedding, our framework can be readily adapted to select high-similarity tokens. From this perspective, we hope to highlight that our method and overall analysis provide insights into token selection for precise pixel grounding, rather than being limited to CLIP-specific properties.
> > >
> > > To this end, we evaluate LiteLVLM on UniPixel$\textemdash$ a recent LVLM built upon the Qwen2.5-VL encoder$\textemdash$to verify its generalizability beyond CLIP-style encoders. We kindly refer the reviewer to our response to Reviewer `Fzx9 (W3 & Q3)`. Consequently, LiteLVLM consistently outperforms LLaVA-PruMerge (ICCV'25) under both 66.7\% and 88.9\% reduction ratios on the Qwen2.5-VL vision encoder (345 visual tokens). These results demonstrate that our method generalizes effectively beyond the CLIP-based vision encoder.
> > >
> > > ---
> > >
> > > **2. Principled and Adaptive Token Selection Ratio.**
> > >
> > > We agree with the reviewer that devising a more principled and adaptive strategy for determining the ratio between similarity-aware and context-aware tokens in LiteLVLM would be a valuable contribution. Recent token pruning methods typically retain visual tokens with predetermined allocations. For instance, when retaining 64 visual tokens, VisionZip uses a fixed allocation of 54 dominant tokens and 10 contextual tokens. In LiteLVLM, when a single referring expression is given, we adopt an empirical 50:50 ratio between similarity-aware and context-aware tokens. However, as similarity-aware tokens are located in the referent region, it is preferable to adapt their number according to the referent size. For example, a larger referent would benefit from allocating more similarity-aware tokens, while a smaller referent requires fewer, allowing more budget for context-aware tokens to capture global context.
> > >
> > > To this end, we devise an adaptive strategy. As the referent size cannot be directly determined at inference time, we estimate the appropriate number of similarity-aware tokens. Specifically, we first sort the visual-text similarity scores in ascending order and select similarities within the Interquartile Range (IQR) bounds, defined by $Q_1-1.5 \cdot IQR$ and $Q_3 + 1.5 \cdot IQR$, where $IQR = Q_3 - Q_1$, to exclude outlier scores. We then identify an "elbow point" as the largest gap between consecutive scores, which indicates the transition between referent-relevant ([REF]) and non-referent tokens. Finally, we select tokens below the elbow point as similarity-aware tokens and fill the remaining budget with context-aware tokens. The table below shows that our adaptive token selection strategy consistently outperforms the fixed 50:50 ratio, improving performance by +0.3 (73.3 vs. 73.6), +1.8 (69.9 vs. 71.7), and +2.9 (63.7 vs. 66.6) under retaining 192, 128, and 64 tokens, respectively, with larger gains observed under higher compression. These results demonstrate that our strategy effectively allocates similarity-aware tokens under a given token budget. We will include these experimental results in the appendix.
> > >
> > > **Table: Ablations for token selection strategy on RefCOCO-*val***
> > > |**Similarity-aware**|**Context-aware**|**RefCOCO**|
> > > |:-:|:-:|:-:|
> > > |**Retain 192 Tokens (&darr; 66.7%)**|||
> > > |50\%|50\%|73.3|
> > > |Adaptive Selection||**73.6**|
> > > |**Retain 128 Tokens (&darr; 77.8%)**|||
> > > |50\%|50\%|69.9|
> > > |Adaptive Selection||**71.7**|
> > > |**Retain 64Tokens (&darr; 88.9%)**|||
> > > |50\%|50\%|63.7|
> > > |Adaptive Selection||**66.6**|
> > >
> > > ---
> > >
> > > We sincerely appreciate your thorough review and valuable suggestions to improve our work. We hope our responses help clarify and address your concerns.

---

### Official Review · Reviewer_UJ5Y · 2026-03-13

**Soundness:** 3
**Presentation:** 4
**Significance:** 3
**Originality:** 2
**Overall Recommendation:** 3
**Confidence:** 5

**Summary:**

This paper explores token pruning methods for large vision-language models, specifically addressing the inefficiencies in pixel grounding tasks caused by redundant visual tokens. The authors propose LiteLVLM, a training-free, text-guided token pruning strategy leveraging an interesting phenomenon they describe as "visual-text similarity reversal." This approach prioritizes visual tokens with low similarity to the [EOS] token for effective grounding and recovers contextual information from tokens with high [CLS] attention scores. Extensive experiments are conducted across multiple datasets to validate the proposed method.

**Compliance With Llm Reviewing Policy:**

Affirmed.

**Final Justification:**

In my discussion with the authors, my primary remaining concern is that the paper largely repackages the known phenomenon of global patch/proxy tokens inherent in CLIP models. The tokens selected by LiteLVLM incidentally overlap with referent regions rather than demonstrating genuine grounding capability. I appreciate the authors' efforts to provide further visualization and clarification regarding this issue. Unfortunately, the updated visualizations have not sufficiently convinced me.

Firstly, the updated visualization replaced the original image examples, significantly weakening the previously observed complementarity between VisionZip and LiteLVLM. The authors have not explained why such a substantial difference in complementarity exists between the first and second visualizations, raising concerns about potentially selective sample choices.

Secondly, regardless of visualization iteration, the set of tokens chosen by LiteLVLM remains diffuse and noisy. I find such token selection inappropriate for grounding tasks that require high positional precision, which reinforces my original review conclusion that LiteLVLM lacks genuine localization capabilities.

Thirdly, the authors have consistently failed to clearly explain why tokens with the lowest similarity to [EOS] are suitable for pixel grounding. They themselves acknowledge in their rebuttal that this approach "is counterintuitive yet effective in practice."

Given these three points, I maintain my original rating.

**Key Questions For Authors:**

1. Both the phenomenon analysis and experimental validation in the paper are based solely on CLIP. However, SigLIP is more commonly used in state-of-the-art LVLMs [1,2]. The authors should verify whether their method remains effective with newer vision encoders (e.g., SigLIP 2 [3]).
2. To clarify the above weaknesses, could the authors provide a visualization comparing the spatial locations of VisionZip’s dominant tokens and LiteLVLM’s similarity-aware tokens under a complementary token count setting, i.e., where the number of dominant tokens plus the number of similarity-aware tokens equals the total number of image patches?

[1] LLaVA-Video: Video Instruction Tuning With Synthetic Data. arXiv 2024.

[2] Qwen3-VL Technical Report. arXiv 2025.

[3] SigLIP 2: Multilingual Vision-Language Encoders with Improved Semantic Understanding, Localization, and Dense Features. arXiv 2025.

**Limitations:**

yes

**Strengths And Weaknesses:**

### Strengths

1. **Originality:** The paper identifies and investigates an intriguing phenomenon, the "visual-text similarity reversal," which provides a fresh perspective on visual-text interactions within CLIP-based models.

2. **Soundness:** The proposed LiteLVLM method is straightforward, training-free, and demonstrates effectiveness in maintaining performance while reducing computational resources significantly.

3. **Presentation:** The overall structure of the paper is clear and well-organized, facilitating easy comprehension and reproduction of results.

### Weaknesses

1. **Questionable Rationale for Text Attention Sink:**
The claim that visual-text similarity reversal stems from the [EOS] token’s lack of semantics feels unconvincing. If [EOS] truly carries little semantic content, how does CLIP achieve strong vision-language alignment? Moreover, why would visual tokens least similar to [EOS] yield accurate pixel grounding? The authors should clarify this contradiction.
2. **Questionable Novelty:**
I wonder whether the paper uncovers a genuinely new phenomenon or is simply repackaging an existing one. Numerous prior works [1,2] have observed a "global patch/proxy token" phenomenon in CLIP: after encoding, image information is largely concentrated in only a few tokens, which often correspond to semantically meaningless background regions. For instance, VisionZip [3] leverages this observation by selecting tokens that receive the highest attention from the [CLS] token as dominant tokens. Furthermore, given CLIP’s contrastive learning objective, the information contained in the [CLS] and  [EOS] tokens should be highly similar.
However, the authors propose using tokens with the lowest similarity to [EOS] for pixel grounding, which essentially selects the complement of VisionZip’s dominant tokens, to avoid focusing on these ineffective regions during grounding. Similarly, the proposed “context-aware tokens” appear analogous to VisionZip’s contextual tokens, serving mainly to complement global information.
3. **Missing Key Citations:**
The paper should include references to and a discussion of papers that have reported similar anomalous distributions of high-information tokens (e.g., DeCLIP [2], DyToK [4]).

Overall, while the visual-text similarity reversal is intriguing and the paper is well-written and logically structured, concerns about the novelty of the core observation prevent me from recommending a higher score at this time. I will adjust my score according to the authors' response.

[1] Explore the Potential of CLIP for Training-Free Open Vocabulary Semantic Segmentation. ECCV 2024.

[2] DeCLIP: Decoupled Learning for Open-Vocabulary Dense Perception. CVPR 2025.

[3] VisionZip: Longer is Better but Not Necessary in Vision Language Models. CVPR 2025.

[4] Less Is More, but Where? Dynamic Token Compression via LLM-Guided Keyframe Prior. NeurIPS 2025.

---

> ### Author Rebuttal · Authors · 2026-03-31
>
> We sincerely thank reviewer `UJ5Y` for the expert feedback and invaluable insights that will help improve our work. Please find our responses to the reviewer's comments below.
>
> ---
>
> > **W1: Questionable Rationale for Text Attention Sink.**
>
> We appreciate the opportunity to clearly elucidate our analysis of the text attention sink. We respectfully emphasize that the attention sink is an active area of research primarily in LLMs [1]. In our work, we identify the text attention sink in CLIP and explain it through the contrastive objective. Specifically, CLIP aligns image-text pair embeddings, which encourages the [EOS] token to align with the [CLS] token. Since the [CLS] encodes global image information, the [EOS] also summarizes the global semantics of the input text. Accordingly, while we agree that [EOS] enables strong global-level vision-language alignment with [CLS], we argue that it carries limited local-level (individual text token) semantics, as its attention is biased toward the [SOS] token. Moreover, prior work [2] highlights the limited region-level understanding in CLIP, which supports our analysis.
>
> We next clarify why visual tokens least similar to [EOS] improve grounding. As detailed above, during pretraining, visual tokens encoding global semantic and background representation receive stronger gradients and become closer to [EOS]. In contrast, [REF] tokens located within localized referent regions receive weaker gradients from [CLS] and remain less similar to [EOS]. For accurate pixel grounding, retaining text-specified [REF] tokens is essential; thus, selecting tokens least similar to [EOS] is counterintuitive yet effective in practice. This is demonstrated in Figure 5 and 7 in the main paper.
>
> [1] *Xiao et al., Efficient Streaming Language Models With Attention Sinks, ICLR, 2024.*
>
> [2] *Zhong et al., RegionCLIP: Region-based Language-Image Pretraining, CVPR, 2022.*
>
> ---
>
> > **W2 & Q2: Questionable Novelty.**
>
> We are grateful for the reviewer’s pointer to CLIP-based works and clarify that our observation is not a repackaging. Unlike prior works, we explicitly utilize textual guidance and analyze visual tokens in relation to the input text, moving beyond [CLS]-centric token concentration. Furthermore, we fully agree with the concern regarding the novelty of our work relative to VisionZip. We acknowledge that the [CLS] and [EOS] tokens are highly similar under CLIP's contrastive objective. However, we contend that there exists a clear modality-dependent difference in the composition of the global information they encode.
>
> To demonstrate this and show that our similarity-aware tokens are not merely the complement of VisionZip's dominant tokens, we present a visualization [(link)](https://anonymous.4open.science/r/CLIP-Tricks-You/rebuttal/_comparison_visionzip.png). Under a complementary token count (288) setting, we quantify the results (Figure 1) and visualize spatial distributions (Figure 2), revealing substantial intersection alongside a notable number of distinct tokens (1-(a)), while similarity-aware tokens are more uniquely concentrated in referent regions (1-(b)). Additionally, while our context-aware tokens and VisionZip's contextual tokens both utilize [CLS] and complement global information, they differ in their scoring mechanism. Specifically, we score tokens using the attention-scaled value vectors for estimating their contribution to [CLS] and perform token selection, whereas VisionZip relies on similarity metrics and performs token merging.
>
> ---
>
> > **W3: Missing Key Citations.**
>
> We thank the reviewer for the comprehensive list of insightful works such as DeCLIP and DyToK, which explore the non-uniform distributions of informative visual tokens. We will add the citations and a comparative discussion in the "Related Work" section of the final version.
>
> ---
>
> > **Q1: Generalization Beyond CLIP.**
>
> To verify the generalization of our method, we analyze visual-text similarity across CLIP-family encoders. Due to the rebuttal length limit, we kindly refer the reviewer to the visualization results in our response to Reviewer `FZx9 (W1-2 & Q1-2)`. Moreover, we evaluate LiteLVLM on X-SAM, a recent LVLM built on the SigLIP-2 encoder, as shown in the table below.
>
> **Table: Performance comparison on X-SAM**
> |**Method**|**RefCOCO**|**RefCOCO+**|**RefCOCOg**|**Avg.**|
> |:-|:-|:-:|:-:|:-:|
> |**X-SAM (AAAI'26)**|85.1|78.0|83.8|82.3|
> |**Retain 244 Tokens (&darr; 66.7%)**|||||
> |VisionZip|78.8|67.5|75.3|73.9|
> |LiteLVLM|79.4|70.8|76.9|**75.7**|
> |**Retain 81 Tokens (&darr; 88.9%)**|||||
> |VisionZip|69.5|51.6|63.8|61.6|
> |LiteLVLM|69.6|54.8|65.9|**63.4**|
>
> - **Discussion:** LiteLVLM consistently outperforms VisionZip across all token budgets. These results demonstrate the robustness of our design and its generalization across diverse LVLMs.
>
> ---
>
> We sincerely appreciate your valuable review and suggestions. If you find our responses satisfactory, we would be grateful for your reconsideration of the rating.

---

> > ### Author Rebuttal · Reviewer_UJ5Y · 2026-04-04
> >
> > Thank you for the detailed clarifications, additional experiments, and data visualizations, which have addressed some of my concerns. However, several points still remain.
> >
> > In Figure 1(b) (provided in your response to W2), could you please explain why the number of intersection tokens is higher than the individual token counts from VisionZip and LiteLVLM? Additionally, the results shown in Figure 1(a) appear quite natural: VisionZip and LiteLVLM primarily select complementary sets of tokens. Even though semantic discrepancies between vision and text modalities might cause some token overlap, it naturally leads to tokens being selected from largely different spatial positions.
> >
> > Moreover, while I sincerely appreciate the author's effort in visualization, Figure 2 indeed confirms my hypothesis that most visual tokens selected by LiteLVLM are complementary to the dominant tokens of VisionZip. The reason LiteLVLM exhibits better grounding performance is likely because VisionZip's token set largely resides in irrelevant background regions. Consequently, the complementary subset chosen by LiteLVLM incidentally overlaps with referent regions, rather than indicating genuine grounding capability. Therefore, my initial concern remains that the paper primarily repackages the known phenomenon of global patch/proxy tokens inherent in CLIP models.

---

> > > ### Author Response · Authors · 2026-04-06
> > >
> > > We sincerely thank the reviewer for the valuable and expert questions, which helped clarify our paper. We also deeply apologize for any confusion and lack of clarity in our rebuttal. Our responses are summarized as below:
> > >
> > > ---
> > >
> > > **1. Clarification on Intersection Token Counts in Figure 1(b).**
> > >
> > > Firstly, we would like to clarify that Figure 1(b) in our response to W2 illustrates the unique token counts for VisionZip (dominant-only) and LiteLVLM (similarity-aware-only), rather than the total number of retained tokens within referent regions ([REF]). We regret that this definition was not explicit, which may have led to the misunderstanding, and note that the total number of retained tokens for each method exceeds the number of intersection tokens, as it is the sum of tokens unique to each method and the intersection tokens. To resolve the confusion, we provide a revised visualization [(link)](https://anonymous.4open.science/r/CLIP-Tricks-You/rebuttal/_comparison_visionzip_rebuttal.png) that presents the total number of retained tokens within referent regions, along with their corresponding numerical counts. We kindly refer the reviewer to this updated visualization.
> > >
> > > ---
> > >
> > > **2. Clarification on the Comparison with VisionZip.**
> > >
> > > - **Figure 1(a) shows substantial overlap, which indicates that LiteLVLM and VisionZip are not strictly complementary.**
> > >
> > > We understand the reviewer's statement that LiteLVLM follows an opposite selection of VisionZip due to the similarity between [CLS] and [EOS] tokens under contrastive learning. However, we respectfully disagree with the reviewer's concern that LiteLVLM's similarity-aware tokens are merely a "complementary" subset of VisionZip's dominant tokens. If LiteLVLM simply preserves a complementary set of tokens to those of VisionZip, the number of intersection tokens would be expected to be relatively small or close to zero, as the tokens selected by each method would largely lie in distinct spatial regions. However, the revised Figure 1(a) clearly shows that the number of intersection tokens exceeds that of both dominant-only and similarity-aware-only tokens. In Figure 1, we follow the reviewer's suggestion and adopt a complementary token count setting, where the numbers of dominant and similarity-aware tokens are each set to 288 (i.e., half of the total 576 image patches). Under this setting, the number of intersection tokens is on average about 15-25 higher than the method-unique token counts and accounts for approximately 26-27\% of the total image patches. Given the substantial overlap observed in the selected tokens, we suggest that LiteLVLM and VisionZip are not strictly complementary.
> > >
> > > - **LiteLVLM adaptively selects tokens depending on the referring expression, indicating that they are not a static complementary subset.**
> > >
> > > Moreover, if LiteLVLM retains a complementary subset of dominant tokens, the retained tokens should remain invariant across different referring expressions for the same image. To verify this, we refer the reviewer to the updated visualization in Figure 2, where different referring expressions correspond to different referents within the same image. As shown in Figure 2, since VisionZip selects dominant tokens based on similarity to the [CLS] token in a text-agnostic manner, these tokens remain anchored to fixed spatial regions regardless of the referring expression (e.g., "blk car" vs. "bottom right car"). In contrast, LiteLVLM exploits the [EOS] token and thus adaptively selects tokens conditioned on the input text, allowing it to better align with the target referent spatially. This also confirms that LiteLVLM adaptively selects its similarity-aware tokens, even when multiple identical objects must be distinguished using spatial or relative cues (e.g., "half umbrella behind the guys' head on the left" vs. "top full umbrella"). From these results, we argue that LiteLVLM's similarity-aware tokens are inherently text-dependent and therefore not consistent with a complementary subset of VisionZip's dominant tokens.
> > >
> > > Furthermore, VisionZip's dominant tokens often reside in background regions, but are also found in visually salient regions. Figure 1(b) shows that a substantial number of dominant tokens lie within the referent regions. However, in Figure 2, we find that VisionZip selects tokens for visually prominent objects such as the "blk car", but fails to retain tokens for less salient ones such as the "bottom right car". In contrast, LiteLVLM effectively retains tokens for the "bottom right car", indicating that it is not an incidental overlap but is guided by the input text. We believe this also clarifies that our method is not a simple repackaging of existing works.
> > >
> > > ---
> > >
> > > We hope our responses rectify the misunderstanding regarding the paper's contribution. If you find our responses satisfactory, we would be grateful for your reconsideration of the rating. We would be happy to address any further questions.

---

### Official Review · Reviewer_FZx9 · 2026-03-15

**Soundness:** 3
**Presentation:** 3
**Significance:** 3
**Originality:** 3
**Overall Recommendation:** 4
**Confidence:** 3

**Summary:**

This paper studies training-free visual token pruning for efficient pixel grounding in large vision-language models. The central observation is a counter-intuitive **visual-text similarity reversal** in CLIP: visual tokens covering the referent region often exhibit **lower**, rather than higher, similarity to the textual representation. Based on this insight, the authors propose `LiteLVLM`, which retains low-similarity visual tokens and further recovers context-aware tokens to preserve global scene information. Experiments on both image and video pixel grounding benchmarks show a favorable trade-off between grounding performance and inference efficiency, with clear gains in speed and memory usage under reduced token budgets.

**Compliance With Llm Reviewing Policy:**

Affirmed.

**Key Questions For Authors:**

1. Does the visual-text similarity reversal still hold when CLIP is replaced with more recent fine-grained visual encoders, such as `SigLIP-2` [1], especially on referring-expression-oriented tasks?

2. How consistent is this phenomenon across different CLIP-family backbones, such as `OpenAI CLIP`, `MetaCLIP`, and `SigLIP`? Is it a general property or mainly an artifact of a specific encoder family?

3. Can the authors evaluate `LiteLVLM` on more recent LVLMs, such as `LLaVA-OneVision` [2] or `Qwen3-VL` [3], to verify whether the pruning rule generalizes beyond the `LLaVA-1.5-7B` setting?

**Limitations:**

Yes

**Strengths And Weaknesses:**

Strengths

1. The paper identifies an inspiring and non-trivial phenomenon in CLIP, namely the **visual-text similarity reversal** of `[REF]` tokens. Both the analytical discussion and the empirical evidence suggest that tokens corresponding to the referent region can have surprisingly low similarity to the referring text, which provides strong motivation for the proposed pruning strategy.

2. The experimental evaluation is extensive. The method is validated not only on image-level pixel grounding but also on video pixel grounding, where it continues to deliver strong accuracy-efficiency trade-offs.

3. The paper includes a rich set of qualitative visualizations. These examples make it easier to see that `LiteLVLM` indeed tends to preserve tokens associated with the referred region during pruning.

4. The efficiency analysis is detailed and practically meaningful. By reporting both inference speed and memory usage, the paper shows that `LiteLVLM` can substantially improve efficiency while preserving most of the original model performance.

5. The proposed pruning strategy appears practically useful, especially because it can be combined with system-level acceleration techniques such as `FlashAttention`, further improving its deployment value.

Weaknesses

1. The paper does not investigate whether the same phenomenon also appears in more recent fine-grained visual encoders commonly used in modern MLLMs. For example, it remains unclear whether encoders such as `SigLIP-2`, which are trained with stronger dense and localization-oriented capabilities, would still exhibit the same reversal behavior.

2. There are many CLIP-family variants, such as `OpenAI CLIP ViT-L/14`, `MetaCLIP ViT-L/14`, and `SigLIP ViT-SO/14`. The paper does not establish whether the observed visual-text similarity reversal is a broadly shared property of CLIP-like encoders or a more model-specific inductive bias.

3. The experiments are conducted only on `LLaVA-1.5-7B`-based systems. It is therefore unclear whether the proposed pruning strategy transfers well to more recent LVLMs, such as `LLaVA-OneVision` [2] or `Qwen3-VL` [3].

4. The title above Figure 2(c) appears to contain a labeling error. It seems that `"Similarity Rank of [REF] Tokens to [CLS]"` should instead be `"Similarity Rank of [REF] Tokens to [EOS]"`.

---

> ### Author Rebuttal · Authors · 2026-03-31
>
> We thank the reviewer `FZx9` for the constructive feedback and for recognizing the non-trivial visual-text similarity reversal. We appreciate your acknowledgment of our extensive evaluation, including qualitative visualizations and efficiency analysis. Our responses to the reviewer's comments are detailed below.
>
> ---
>
> > **W1-2 & Q1-2: Visual-Text Similarity Reversal in Modern Visual Encoders.**
>
> We appreciate the reviewer's suggestion to investigate more recent fine-grained visual encoders and CLIP-family variants. As shown in the Figure [(link)](https://anonymous.4open.science/r/CLIP-Tricks-You/rebuttal/comparison_encoder.png), we visualize the [REF]-[EOS] similarity rank distributions across diverse vision-language encoders: (a) MetaCLIP and (b) SigLIP still hold the visual-text similarity reversal, where referent tokens tend to have lower similarity, while (c) SigLIP-2 shows a more centered distribution. From these results, we observe that SigLIP-2, which introduces additional localization and dense prediction losses (LocCa and SILC/TIPS), encourages finer token-level alignment, alleviating the reversal effect. These findings suggest that the similarity reversal arises from the global image-text alignment objective and is broadly shared across CLIP-like encoders widely used in LVLMs (*e.g.,* LLaVA, LLaVA-OneVision). We respectfully highlight that LiteLVLM is explicitly designed to leverage this property for effective token pruning.
>
> ---
>
> > **W3 & Q3: Extending LiteLVLM to Recent LVLM.**
>
> We agree with the reviewer that our experiments are conducted on LLaVA-1.5-7B-based systems. To validate the generalization of our method, we conduct experiments on UniPixel [1], built upon Qwen2.5-VL (345-token setting). Following VisionZip and VisPruner, we use the average attention over patch tokens as a proxy for the [CLS] token and compare with LLaVA-PruMerge on the RefCOCO/+/g *val* subsets. The table below shows that LiteLVLM outperforms LLaVA-PruMerge by +3.8 and +2.3 when retaining 114 and 78 tokens, respectively, demonstrating strong generalization of our token pruning method. We also evaluate LiteLVLM on a model based on SigLIP-2 and Phi-3-3.8B, and kindly refer the reviewer to our response to Reviewer `UJ5Y (Q1)`.
>
> **Table: Generalization of LiteLVLM on Qwen2.5-VL-based LVLM under different token budgets**
> |**Method**|**LVLM**|**RefCOCO**|**RefCOCO+**|**RefCOCOg**|**Avg.**|
> |:-|:-|:-:|:-:|:-:|:-:|
> |**UniPixel-3B**|Qwen2.5-VL|80.5|74.3|76.3|77.0|
> |**Retain 114 Tokens (&darr; 66.7%)**||||||
> |LLaVA-PruMerge (ICCV'25)||75.2|65.8|70.8|70.6 (-6.4)|
> |LiteLVLM||79.2|68.3|75.6|**74.4 (-2.6)**|
> |**Retain 78 Tokens (&darr; 88.9%)**||||||
> |LLaVA-PruMerge (ICCV'25)||72.1|60.9|66.6|66.5 (-10.5)|
> |LiteLVLM||74.1|62.1|70.1|**68.8 (-8.2)**|
>
> [1] *Liu et al., UniPixel: Unified Object Referring and Segmentation for Pixel-Level Visual Reasoning, NeurIPS, 2025.*
>
> ---
>
> > **W4: Figure 2 Labeling Error.**
>
> We thank the reviewer for catching this issue that we missed. We will update the [figure](https://anonymous.4open.science/r/CLIP-Tricks-You/rebuttal/figure2_rebuttal.png) in the final version.
>
> ---
>
> We sincerely appreciate your valuable review and suggestions. If you find our responses satisfactory, we would be grateful for your reconsideration of the rating.

---

### Decision · Program_Chairs · 2026-04-30

**Decision:**

Accept (regular)

**Comment:**

This paper was reviewed by four experts, resulting in three Weak Accepts and one Weak Reject. Reviewer UJ5Y maintained a Weak Reject, arguing that the method might be "repackaging" the known phenomenon of CLIP proxy tokens. The reviewer expressed concern that the token selection was "diffuse and noisy" and questioned the conceptual rationale behind the similarity reversal.

The AC read the paper carefully. The paper identifies a counterintuitive "visual-text similarity reversal" in CLIP, where visual tokens corresponding to referent regions exhibit lower similarity to textual representations than background tokens. Leveraging this, the authors propose LiteLVLM, a training-free token pruning strategy for efficient pixel grounding.

The AC finds the discovery in the paper interesting and deserves to be seen and discussed by a broader audience, and also acknowledges that explaining the reason could be difficult. The experimental results are strong. Therefore, the AC overweighs the paper’s original discovery and empirical contributions over the concerns and recommends for acceptance. The authors are encouraged to try their best to improve the paper and incorporate the discussions in the rebuttal into the final version.